# Learning What Matters: Steering Diffusion via Spectrally Anisotropic Forward Noise

## Abstract

Diffusion Probabilistic Models (DPMs) have achieved strong generative performance, yet their inductive biases remain largely implicit. In this work, we aim to build inductive biases into the training and sampling of diffusion models to better accommodate the target distribution of the data to model. We introduce an anisotropic noise operator that shapes these biases by replacing the isotropic forward covariance with a structured, frequency-diagonal covariance. This operator unifies band-pass masks and power-law weightings, allowing us to emphasize or suppress designated frequency bands, while keeping the forward process Gaussian. We refer to this as Spectrally Anisotropic Gaussian Diffusion (SAGD). In this work, we derive the score relation for anisotropic forward covariances and show that, under full support, the learned score converges to the true data score as $t \to 0$, while anisotropy reshapes the probability-flow path from noise to data. Empirically, we show the induced anisotropy outperforms standard diffusion across several vision datasets, and enables *selective omission*: learning while ignoring known corruptions confined to specific bands. Together, these results demonstrate that carefully designed anisotropic forward noise provides a simple, yet principled, handle to tailor inductive bias in DPMs.

## 1 Introduction

Diffusion Probabilistic Models (DPMs) have emerged as powerful tools for approximating complex data distributions, finding applications across a variety of domains, from image synthesis to probabilistic modeling (Yang et al., 2024; Ho et al., 2020; Sohl-Dickstein et al., 2015; Venkatraman et al., 2024; Sendera et al., 2024). These models operate by gradually transforming data into noise through a defined diffusion process and training a denoising model (Vincent et al., 2008; Alain & Bengio, 2014) to learn to reverse this process, enabling the generation of samples from the desired distribution via appropriate scheduling. Despite their success, the inductive biases inherent in diffusion models remain largely unexplored, particularly in how these biases influence model performance and the types of distributions that can be effectively modeled.

Inductive biases are known to play a crucial role in deep learning models, guiding the learning process by favoring certain types of data representations over others (Geirhos et al., 2019; Bietti & Mairal, 2019; Tishby & Zaslavsky, 2015). A well-studied example is the Frequency Principle (F-principle) or spectral bias, which suggests that neural networks tend to learn low-frequency components of data before high-frequency ones (Xu et al., 2019; Rahaman et al., 2019). Another related phenomenon is what is also known as the simplicity bias, or shortcut learning (Geirhos et al., 2020; Scimeca et al., 2021; 2023b), in which models are observed to preferentially pick up on simple, easy-to-learn, and often spuriously correlated features in the data for prediction. If left implicit, it is often unclear whether these biases will improve or hurt the models' performance on downstream tasks, potentially leading to undesired outcomes (Scimeca et al., 2023a). In this work, we aim to explicitly tailor the inductive biases of DPMs to better learn the target distribution of interest.

Recent studies have begun to explore the inductive biases inherent in diffusion models. For instance, Kadkhodaie et al. (2023) analyzes how the inductive biases of deep neural networks trained for image denoising contribute to the generalization capabilities of diffusion models. They demonstrate that these biases lead to geometry-adaptive harmonic representations, which play a crucial role in the models' ability to generalize beyond the training data. Similarly, Zhang et al. (2024) investigates

the role of inductive and primacy biases in diffusion models, particularly in the context of reward optimization. They propose methods to mitigate overoptimization by aligning the models' inductive biases with desired outcomes. Other methods, such as noise schedule adaptations (Sahoo et al., 2024) and the introduction of non-Gaussian noise (Bansal et al., 2022) have shown promise in improving the performance of diffusion models on various tasks. However, the exploration of frequency domain techniques within diffusion models is a relatively new area of interest. One of the pioneering studies in this domain investigates the application of diffusion models to time series data, where frequency domain methods have shown potential for capturing temporal dependencies more effectively (Crabbé et al., 2024). Similarly, the integration of spatial frequency components into the denoising process has been explored for enhancing image generation tasks (Qian et al., 2024; Yuan et al., 2023), showcasing the importance of considering frequency-based techniques as a means of refining the inductive biases of diffusion models.

In this work, we explore a new avenue, to build inductive biases in DPMs by frequency-based noise control. Specifically, we replace the isotropic forward noise with an anisotropic forward Gaussian operator whose covariance is structured in the Fourier basis. The main hypothesis in this paper is that the noising operator in a diffusion model has a direct influence on the model's representation of the data. Intuitively, the information erased by the noising process is the very information that the denoising model has pressure to learn, so that reconstruction is possible. Accordingly, by shaping the forward covariance we can steer which modes carry supervision signal during training and thus which aspects of the data distribution the model learns most effectively. We focus our attention on the generative learning of topologically structured data, and implement anisotropy via a frequency-parameterized schedule that emphasizes or de-emphasizes selected bands while keeping the forward process Gaussian. In what follows we refer to this setting as Spectrally Anisotropic Gaussian Diffusion (SAGD).

We report several key findings showing that SAGD provides a simple, principled handle to tailor inductive biases, including: improved learning of information lying in particular frequency bands; increased performance across several natural image datasets; and learning while ignoring (corrupted) information at predetermined frequency bands. Because SAGD modifies only the forward covariance, it integrates with existing diffusion implementations with a few lines of code and preserves the rest of the pipeline intact. We summarize our contributions as follows:

1. We introduce SAGD, an anisotropic forward-noise operator with frequency-diagonal covariance (in the Fourier basis) that provides a simple handle on spectral inductive bias.

2. We provide a theoretical analysis showing that, under full spectral support, the learned score at $t \to 0$ recovers the true data score, while anisotropy deterministically reshapes the probability-flow path.

3. We show that SAGD can steer models to better approximate information concentrated in selected bands of the underlying data distribution.

4. We test and empirically show that models trained with SAGD anisotropic forward covariance can match or outperform traditional (isotropic) diffusion across multiple datasets.

5. We demonstrate *selective omission*: by zeroing chosen bands in the forward covariance, models learn to ignore known corruptions and recover the clean distribution.

## 2 METHODS

### 2.1 DENOISING PROBABILISTIC MODELS (DPMs)

Denoising Probabilistic Models are a class of generative models that learn to reconstruct complex data distributions by reversing a gradual noising process. DPMs are characterized by a *forward* and *backward* process. The *forward process* defines how data is corrupted, typically by Gaussian noise, over time. Given a data point $\mathbf{x}_0$ sampled from the data distribution $q(\mathbf{x}_0)$, the noisy versions of the data $\mathbf{x}_1, \mathbf{x}_2, \ldots, \mathbf{x}_T$ are generated according to:

$$q(\mathbf{x}_t \mid \mathbf{x}_{t-1}) = \mathcal{N}(\mathbf{x}_t; \sqrt{\alpha_t}\, \mathbf{x}_{t-1}, (1 - \alpha_t)\, \mathbf{I}), \tag{1}$$

with variance schedule $\alpha_t$. The *reverse process* models the denoising operation, attempting to recover $\mathbf{x}_{t-1}$ from $\mathbf{x}_t$:

$$p_\theta(\mathbf{x}_{t-1} \mid \mathbf{x}_t) = \mathcal{N}(\mathbf{x}_{t-1}; \mu_\theta(\mathbf{x}_t, t), \sigma_t^2 \mathbf{I}), \tag{2}$$

where $\mu_\theta(\mathbf{x}_t, t)$ is predicted by a neural network $f_\theta$, and the variance $\sigma_t^2$ can be fixed, learned, or precomputed based on a schedule. We train the denoising model with the standard $\epsilon$–parameterization by minimizing

$$\mathcal{L} = \mathbb{E}_{t, \mathbf{x}_0, \epsilon \sim \mathcal{N}(0,\mathbf{I})} \left[ \left\| \epsilon - \epsilon_\theta(\mathbf{x}_t, t) \right\|^2 \right], \qquad \mathbf{x}_t = \sqrt{\bar{\alpha}_t}\, \mathbf{x}_0 + \sigma_t\, \epsilon, \tag{3}$$

where $\bar{\alpha}_t = \prod_{s=1}^t \alpha_s$ and $\sigma_t^2 = 1 - \bar{\alpha}_t$, $\epsilon$ is the Gaussian noise added to $\mathbf{x}_0$, and $\epsilon_\theta$ is the model's prediction of this noise. To generate new samples, we start from noise and apply the learned reverse process iteratively.

## 2.2 Spectrally Anisotropic Gaussian Diffusion (SAGD)

The objective of this section is to convolve the data with a spectrally anisotropic gaussian noise during training to steer a model's tendency to learn particular aspects of the data distribution. To do so, we wish to generate spatial Gaussian noise whose frequency content can be systematically manipulated according to an arbitrary weighting function in the Fourier basis, where stationary Gaussian covariances diagonalize and our power-law and band-pass families become low-dimensional parameterizations of the covariance eigenvalues. The right-hand side of Equation 3 denotes how $\mathbf{x}t$ is generated by adding Gaussian noise $\epsilon \sim \mathcal{N}(0, \mathbf{I})$ to $\mathbf{x}0$.

Let us denote by $\mathbf{x} \in \mathbb{R}^{H \times W}$ an image (or noise field) in the spatial domain, and by $\mathcal{F}$ the two-dimensional Fourier transform operator. Given white spatial Gaussian noise $\epsilon \sim \mathcal{N}(0, \mathbf{I})$, we form its Fourier transform $\mathbf{N}_{\mathrm{freq}} = \mathcal{F}(\epsilon)$, where $\mathbf{N}_{\mathrm{freq}} \in \mathbb{C}^{H \times W}$ is a complex-valued random field whose real and imaginary parts are i.i.d. Gaussian, i.e.:

$$\mathbf{N}_{\mathrm{freq}} = \mathbf{N}_{\mathrm{real}} + i\,\mathbf{N}_{\mathrm{imag}}, \quad \mathbf{N}_{\mathrm{real}}, \mathbf{N}_{\mathrm{imag}} \sim \mathcal{N}(0, \mathbf{I}), \tag{4}$$

We introduce a *weighting function* $w(f_x, f_y)$ that scales the amplitude of each frequency component. Let $\mathbf{f} = (f_x, f_y)$ denote coordinates in frequency space, where $f_x = \frac{k_x}{W}$, $f_y = \frac{k_y}{H}$, and $k_x, k_y$ are integer indices (ranging over the width and height), while $H$ and $W$ are the image dimensions. We define the frequency-controlled noise $\mathbf{N}_{\mathrm{freq}}^{(w)}(\mathbf{f})$ as:

$$\mathbf{N}_{\mathrm{freq}}^{(w)}(\mathbf{f}) = \mathbf{N}_{\mathrm{freq}}(\mathbf{f}) \odot w(\mathbf{f}), \tag{5}$$

After applying $w(\mathbf{f})$ in the frequency domain, we invert back to the spatial domain to obtain the noise $\epsilon^{(w)}$:

$$\epsilon^{(w)} = \mathfrak{R}\left( \mathcal{F}^{-1}\big(\mathbf{N}_{\mathrm{freq}}^{(w)}\big) \right), \tag{6}$$

where $\mathfrak{R}(\cdot)$ ensures that our final noise field is purely real.[1]

In practice, any standard spatial noise can be converted to $\epsilon^{(w)}$ via this unified framework:

$$\epsilon \xrightarrow{\mathcal{F}} \mathbf{N}_{\mathrm{freq}} \xrightarrow{w(\mathbf{f})} \mathbf{N}_{\mathrm{freq}}^{(w)} \xrightarrow{\mathcal{F}^{-1}} \epsilon^{(w)}.$$

Note that standard white Gaussian noise is a special case of this formulation, where $w(\mathbf{f}) = 1$ for all $\mathbf{f}$. In contrast, more sophisticated weightings allow one to emphasize, de-emphasize, or even remove specific bands of the frequency domain.

**Theoretical consistency** We keep the forward process Gaussian while reshaping its spectrum. Let $\mathcal{F}$ be the unitary DFT and $w(\mathbf{f}) \geq 0$ a fixed spectral weight. The linear map $\mathbf{T}_w = \mathcal{F}^{-1}\mathrm{Diag}(w)\mathcal{F}$ sends white noise to frequency-based noise $\epsilon^{(w)} = \mathbf{T}_w\,\epsilon$ with covariance

$$\Sigma_w = \mathbf{T}_w\,\mathbf{T}_w^* = \mathcal{F}^{-1}\mathrm{Diag}\big(|w(\mathbf{f})|^2\big)\,\mathcal{F}. \tag{7}$$

Replacing $\epsilon$ by $\epsilon^{(w)}$ in the forward step yields the marginal

$$q_w(\mathbf{x}_t \mid \mathbf{x}_0) = \mathcal{N}\big(\sqrt{\bar{\alpha}_t}\,\mathbf{x}_0, \sigma_t^2\,\Sigma_w\big). \tag{8}$$

---

[1] Since the DFT of a real signal has Hermitian symmetry, multiplying by a real, pointwise weight $w$ preserves Hermitian symmetry and yields a real-valued inverse transform.

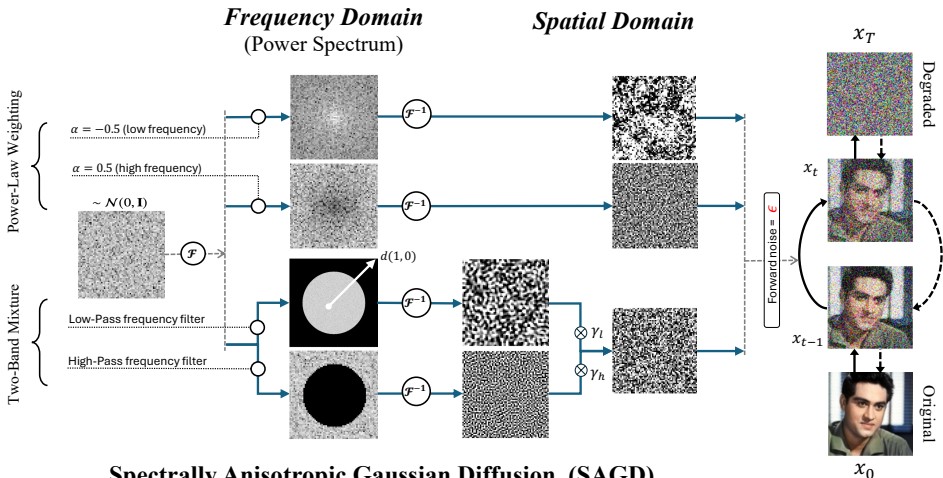

Figure 1: Spectrally Anisotropic Gaussian Diffusion under a generalized framework.

Training with the standard $\ell_2$ objective on the added noise remains optimal: $\epsilon_\theta^\star(\mathbf{x}_t, t) = \mathbb{E}[\epsilon^{(w)} \mid \mathbf{x}_t]$, and the corresponding score satisfies

$$\nabla_{\mathbf{x}_t} \log q_{w,t}(\mathbf{x}_t) \;=\; -\frac{1}{\sigma_t} \Sigma_w^{-1} \epsilon_\theta^\star(\mathbf{x}_t, t), \tag{9}$$

so converting $\epsilon$-predictions to scores simply multiplies by $\Sigma_w^{-1}$. As $t \to 0$, if $\Sigma_w \succ 0$ and $q$ has a locally positive density with $\nabla \log q \in L^1_{\mathrm{loc}}$, the anisotropic Gaussian smoothing collapses to a Dirac and $\nabla \log q_{w,t} \to \nabla \log q$ almost everywhere. Thus, shaping the forward spectrum preserves the endpoint score while altering the path to the score at the data distribution (see Appendix subsection D.1 for proofs and extensions).

## 2.3 FREQUENCY NOISE OPERATORS

In this work, the design of $w(\mathbf{f})$ is especially important. We propose two particular choices which provide a flexible design bench: power-law weighting and a two-band mixture.

### POWER-LAW WEIGHTING (*plw*-SAGD)

We implement a radial power-law anisotropic noise operator that imposes a linear slope in the log–log power spectrum. Let $\mathbf{f} = (f_x, f_y)$ denote normalized frequency coordinates on $[-\frac{1}{2}, \frac{1}{2}]^2$, and define the radial frequency

$$r(\mathbf{f}) \;=\; \sqrt{f_x^2 + f_y^2}.$$

Given white spatial Gaussian noise $\epsilon \sim \mathcal{N}(0, \mathbf{I})$, we form its Fourier transform $\mathbf{N}_{\mathrm{freq}} = \mathcal{F}(\epsilon)$ and scale each frequency bin by

$$w_\alpha(\mathbf{f}) \;=\; (r(\mathbf{f}) + \varepsilon)^\alpha, \qquad \varepsilon = 10^{-10}, \tag{10}$$

where $\alpha \in \mathbb{R}$ controls the slope and $\varepsilon$ is a small weight to prevent a DC singularity. The shaped spectrum and spatial noise are

$$\mathbf{N}_{\mathrm{freq}}^{(\alpha)}(\mathbf{f}) \;=\; \mathbf{N}_{\mathrm{freq}}(\mathbf{f}) \cdot w_\alpha(\mathbf{f}), \qquad \epsilon^{(\alpha)} \;=\; \Re(\mathcal{F}^{-1}[\mathbf{N}_{\mathrm{freq}}^{(\alpha)}]), \tag{11}$$

which we use in the forward step $\mathbf{x}_t = \sqrt{\alpha_t}\, \mathbf{x}_{t-1} + \sqrt{1 - \alpha_t}\, \epsilon^{(\alpha)}$. A minimal code implementation of *plw*-SAGD can be found in Appendix E.

**Effect on spectrum and learning signal.** Let $w_\alpha(r) = (r + \varepsilon)^\alpha$ be the radial spectral weight. Because amplitudes are scaled by $w_\alpha$ while power scales with $|w_\alpha|^2$, the radially averaged power spectral density (RAPSD) follows: $\log \mathrm{PSD}(r) \approx (2\alpha) \log r + \mathrm{const}$, so $\alpha > 0$ tilts energy toward

high frequencies (sharper textures), $\alpha < 0$ toward low frequencies (coarser structure), and $\alpha = 0$ recovers white noise. Note also that the global scalar rescaling of $\Sigma_w$ can be absorbed into the scalar variance $\sigma_t^2$ in the forward process (or the noise schedule). What we aim to do, instead, is the relative weighting of the modes (eigenvalues) of $\Sigma_w$ in the Fourier basis, (shape of $w(\mathbf{f})$).

BAND-PASS MASKING AND TWO-BAND MIXTURE (*bpm*-SAGD))

A band-pass mask can be viewed as a special case of a more general weighting function:

$$w(\mathbf{f}) \in \{0, 1\}. \tag{12}$$

In this case, the frequency domain is split into a set of permitted and excluded regions, or radial thresholds. With this, we can construct several types of filters, including a low-pass filter retaining only frequencies below a cutoff (e.g., $\|\mathbf{f}\| \leq \omega_c$), a high-pass filter keeping only frequencies above a cutoff, or more generally a filter restricting $\|\mathbf{f}\|$ to lie between two thresholds $[a, b]$. We thus define a simple band-pass filter as:

$$w_{a,b}(\mathbf{f}) = \mathbf{M}_{[a,b]}(f_x, f_y) = \begin{cases} 1, & \text{if } a \leq d(f_x, f_y) \leq b, \\ 0, & \text{otherwise,} \end{cases} \tag{13}$$

In this special case, $w(\mathbf{f})$ is simply a binary mask, selecting only those frequencies within $[a, b]$.

For the experiments in this paper we formulate a simple two-band mixture, where we limit ourselves to constructing noise as a linear combination of two band-pass filtered components. Specifically, we generate frequency-filtered noise $\epsilon_f$ via:

$$\epsilon^{(w)} = \gamma_l \, \epsilon_{[a_l, b_l]} + \gamma_h \, \epsilon_{[a_h, b_h]}, \tag{14}$$

where $\gamma_l, \gamma_h \geq 0$ denote the relative contributions of a low and a high-frequency component ($\gamma_l + \gamma_h = 1$), each filtering noise respectively in the ranges $[a_l, b_l]$ (low-frequency range) and $[a_h, b_h]$ (high-frequency range). We uniquely refer to $\epsilon_{[a,b]}$ as the noise filtered in the $[a, b]$ frequency range following Equation 5 and Equation 6. Standard Gaussian noise emerges as a particular instance of this formulation, for $\gamma_l = 0.5$, $\gamma_h = 0.5$, $a_l = 0$, $b_l = a_h = 0.5$, and $b_h = 1$. We provide a minimal code implementation of *bpm*-SAGD in Appendix E.

**Selective omission:** If $w$ vanishes on a band, then $\Sigma_w$ is rank-deficient and the model learns the score *projected* onto range($\Sigma_w$). In the two-band mixture operator, we can achieve this for $b_l < a_h$, leaving the $[b_l, a_h]$ frequency band unsupported by the anisotropic covariance. Note that the $t \to 0$ score-consistency result in our analysis requires full spectral support ($\Sigma_w \succ 0$); when $\Sigma_w$ is singular, the smoothed marginals are not absolutely continuous and the estimator converges only to the *projected* score, i.e., $\Pi \nabla_{\mathbf{x}} \log q(\mathbf{x})$ with $\Pi$ the orthogonal projector onto range($\Sigma_w$) (equivalently, replace $\Sigma_w^{-1}$ by the Moore–Penrose pseudoinverse in the score–$\epsilon$ mapping). As later shown, we exploit this to avoid learning components in the omitted band, while learning and recovering the information in the bands of interest.

## 3 RESULTS

### 3.1 EXPERIMENTAL DETAILS

All experiments involve separately training and testing DPMs with various SAGD schedules, alongside standard isotropic Gaussian baselines. We consider six image datasets—MNIST, CIFAR-10, DomainNet-Quickdraw, WikiArt, FFHQ, and ImageNet-1k—spanning widely different visual distributions, scales, and statistics. We study both pixel-space diffusion with U-Net denoisers and latent diffusion with DiT backbones in a DINOv2 feature space; the latter uses the public RAE implementation of state-of-the-art DiT models on ImageNet-1k at 256×256 resolution (Zheng et al., 2025). We use DDIM sampling (Song et al., 2021) in all experiments, so no step noise is injected at test time. As quality metrics, we report FID and KID between generated samples and held-out data, computed from Inception-v3 features: for all datasets except ImageNet-1k we use a 768-dimensional intermediate block, while for ImageNet-1k we follow standard practice and use the 2048-dimensional penultimate block. Unless otherwise stated, we report averages over multiple random seeds. Additional experimental details are provided in Appendix C.

## 3.2 IMPROVED SAMPLING VIA SAGD

In the first set of experiments, we wish to test our main hypothesis, i.e. that appropriate spectral manipulations of the forward noise can better support the learning of particular aspects of the sampling distribution. In the following experiments, we use the *plw*-SAGD formulation in Equation 10 and Equation 11 to train and compare diffusion models with different anisotropic power-law weighted operators, while varying the value of $\alpha$ to emphasize or de-emphasize the learning of higher or lower frequency components of the target distribution, where $\alpha \in [-0.1, 0.1]$.

### 3.2.1 QUALITATIVE OVERVIEW

First, we show a qualitative example of a standard forward linear noising schedule in DPMs, as compared to two particular settings of *plw*-SAGD, emphasizing high and low-frequency noising in Figure 2. With standard noise, information is uniformly removed from the image, with sample quality degrading evenly over time. In the high-frequency noising schedule ($\alpha > 0$), sharpness and texture are affected more significantly than general contours and shapes, which remain intact over longer trajectories; in the low-frequency noising schedule ($\alpha < 0$), instead, general shapes and homogeneous pixel clusters are quickly affected, yielding qualitatively different information destruction operations over the sampling time steps. As discussed previously, we hypothesize that this will, in turn, purposely affect the information learned by the denoiser, effectively focusing the diffusion sampling process on different parts of the distribution to learn.

### 3.2.2 LEARNING TARGET DISTRIBUTIONS FROM FREQUENCY-BOUNDED INFORMATION

We conduct a controlled experiment to test whether SAGD yields better samplers in the case where the information content in the data lies, by construction, in the high frequencies. We use the CIFAR-10 dataset, and corrupt the original data with low-frequency noise $\epsilon_{[0,.3]}$, thus erasing the low-frequency content while predominantly preserving the high-frequency details in the range $\epsilon_{[.3,1]}$. We separately train DPMs with twelve different $\alpha$ values, as well as a standard DPM at $\alpha = 0$ (*baseline*). We repeat the experiment over three seeds and report the average FID and error in Figure 3. In the figure, we observe an almost monotonically decreasing relationship between the mean FID and increasing values of $\alpha$, with a significant 0.3 decrease in average FID score for $\alpha = 0.08$ from the standard DPM baseline. The observation is in line with our original intuition, whereby improved learning can be achieved by aligning the forward noising operation with the data's dominant spectral content (the frequency bands carrying most of the information).

### 3.2.3 SAGD IN NATURAL DATASETS

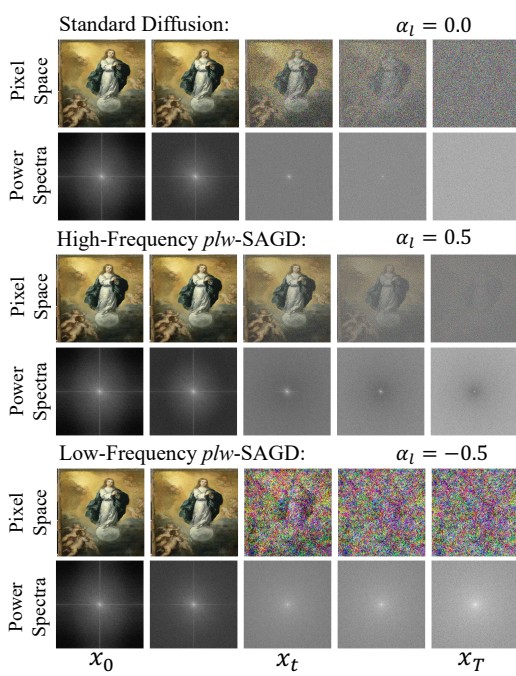

Figure 2: Power spectra and image visuals of the forward Process in standard diffusion, as compared to high ($\alpha = 0.5$) and low-frequency ($\alpha = -0.5$) noise settings of a power-law weighted SAGD.

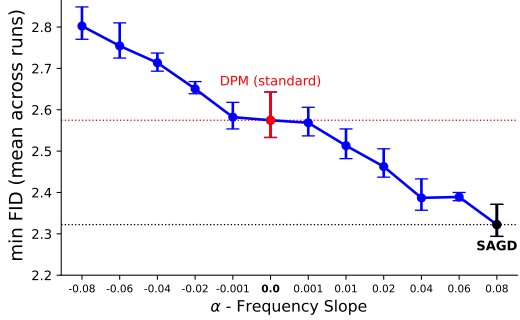

Figure 3: Mean FID across seeds of *plw*-SAGD diffusion samplers trained on CIFAR-10, pre-processed to only retain high frequency information ($\alpha = 0$ yields standard diffusion).

Table 1: FID across selected $\alpha$ (frequency slope) settings for all datasets (mean ± standard error across seeds). FID is computed from block 768 for all datasets except ImageNet1k (block 2048).

| Dataset → | MNIST | CIFAR-10 | Domainnet-Quickdraw | Wiki-Art | FFHQ | ImageNet1k |
|---|---|---|---|---|---|---|
| baseline | $0.42_{\pm 8.52e-03}$ | $0.75_{\pm 0.01}$ | $0.60_{\pm 0.05}$ | $1.06_{\pm 0.03}$ | $1.11_{\pm 0.01}$ | $8.6819_{\pm 0.0739}$ |
| $\alpha = -0.060$ | $0.28_{\pm 0.02}$ | $0.94_{\pm 0.02}$ | $0.52_{\pm 0.05}$ | $1.35_{\pm 0.08}$ | $1.74_{\pm 0.10}$ | $8.1098_{\pm 0.0229}$ |
| $\alpha = -0.040$ | $0.31_{\pm 7.76e-03}$ | $0.86_{\pm 0.02}$ | $0.49_{\pm 0.02}$ | $1.25_{\pm 0.07}$ | $1.76_{\pm 0.08}$ | $7.5534_{\pm 0.0556}$ |
| $\alpha = -0.020$ | $0.37_{\pm 6.36e-03}$ | $0.76_{\pm 0.01}$ | $0.52_{\pm 0.03}$ | $1.14_{\pm 0.05}$ | $1.68_{\pm 0.19}$ | $7.6419_{\pm 0.0581}$ |
| $\alpha = -0.010$ | $0.37_{\pm 0.02}$ | $0.75_{\pm 0.01}$ | $0.54_{\pm 0.04}$ | $1.09_{\pm 0.04}$ | $1.48_{\pm 0.12}$ | $8.0400_{\pm 0.0236}$ |
| $\alpha = -0.001$ | $0.40_{\pm 0.02}$ | $0.76_{\pm 0.01}$ | $0.56_{\pm 0.04}$ | $1.02_{\pm 5.66e-03}$ | $1.04_{\pm 5.17e-03}$ | $8.5288_{\pm 0.0112}$ |
| $\alpha = 0.010$ | $0.43_{\pm 0.02}$ | $0.80_{\pm 0.02}$ | $0.66_{\pm 0.02}$ | $1.20_{\pm 0.07}$ | $2.06_{\pm 0.06}$ | $9.3867_{\pm 0.0348}$ |

We further test our hypothesis by training twelve SAGD noising models with $\alpha \in [-0.08, 0.08]$ against a standard DPM baseline for each of the datasets considered in our experiments, varying greatly in size, resolution, visual distribution, and complexity, namely: MNIST, CIFAR-10, Domainnet-Quickdraw, Wiki-Art, FFHQ and ImageNet-1k. We report additional information on the datasets and our pre-processing pipeline in Appendix B and Appendix C. We repeat the experiments over 3 seeds and present a focused report of the mean FID and KID metrics for all ablations in Table 1 (see additional results in Appendix F). In Table 1, we observe significant SAGD improvements over standard DPM training (first row) in almost all datasets, where in 5/6 we achieve substantially improved baseline FID scores.

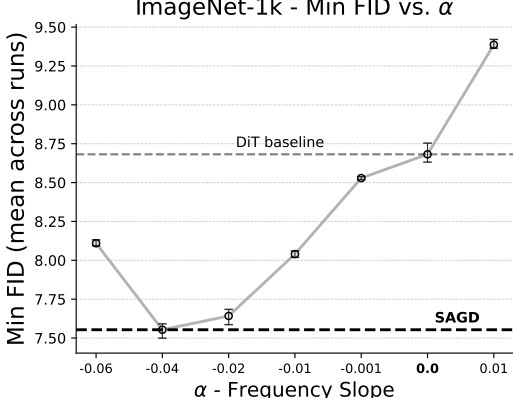

Figure 4: Mean FID across seeds of *plw*-SAGD diffusion samplers trained on ImageNet1k ($\alpha = 0$ yields standard diffusion).

On ImageNet-1k at 256×256 resolution, using the RAE DiT backbone in DINOv2 latent space, the isotropic baseline ($\alpha$=0) attains an FID of 8.68 ± 0.07, whereas SAGD with a low-frequency tilt ($\alpha$=−0.04) reaches 7.55 ± 0.06, i.e., an absolute improvement of ≈ 1.1 FID (about 13% relative). As summarized in Table 1 and visualized in Figure 4, FID decreases almost monotonically as $\alpha$ moves from 0 toward moderately negative values (down to around −0.04) and then worsens again for more negative (−0.06) and positive values of $\alpha$, indicating a non-trivial optimum away from the standard Gaussian setting. This demonstrates that SAGD yields measurable gains even on large-scale, high-resolution, natural-image generation in a state-of-the-art latent DiT setup, and that its benefits are not limited to small or low-resolution datasets.

### 3.3 Selective Learning: Noise Control to Omit Targeted Information

Following our original intuition, learning pressure in DPMs is aligned with the information deletion induced by forward noising. Conversely, when the noising operator is crafted to leave parts of the original distribution intact, no such pressure exists, and the sampler can effectively discard the left-out information during generation.

In this section, we perform experiments whereby the original data is corrupted with noise at different frequency ranges. The objective is to manipulate the inductive biases of diffusion samplers to avoid learning the corruption noise, while correctly approximating the relevant information in the data. We formulate our corruption process as $\mathbf{x}' = A_c(\mathbf{x})$, where $A_c(\mathbf{x}) = \mathbf{x} + \gamma_c \epsilon_{f[a_c, b_c]}$ and $\epsilon_{[a_c, b_c]}$ denotes noise in the $[a_c, b_c]$ frequency range. We perform the corruption on the fly, and use the original MNIST dataset for training while testing on 10K images. We default $\gamma_c = 1$. and show samples of the original and corrupted distributions in Figure 5. For any standard DPM training procedure, as expected, the sampler learns the corrupted distribution presented at training time. As such, the recovery of the original, noiseless, distribution would normally be impossible. Assuming knowledge of the corruption process, we use our two-band mixture formulation in Equation 14 and frame the frequency diffusion learning procedures as a noiseless distribution recovery process, where $a_l = 0$, $b_h = 1$, $b_l = a_c$, and $a_h = b_c$. This formulation effectively allows for the forward frequency-noising operator to omit the range of frequencies in which the corruption lies. In line with our previous

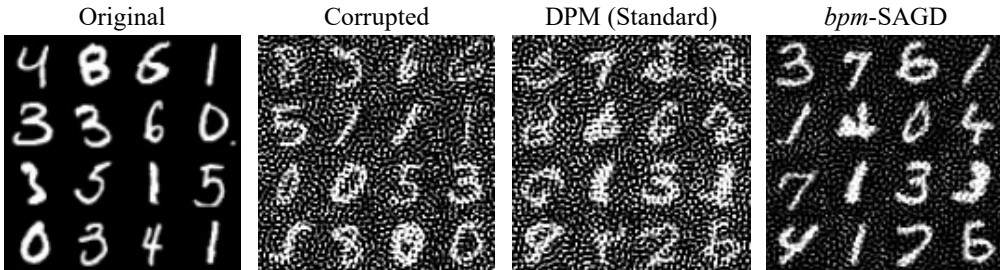

| Original | Corrupted | DPM (Standard) | *bpm*-SAGD |

Figure 5: Samples from the original data distribution, the degraded data distribution, a standard diffusion sampler trained on the degraded data distribution, and a *frequency diffusion* sampler trained on the degraded data distribution. We generate noise for data corruption in the frequency range $[a_c = 0.4, b_c = 0.5)$].

rationale, this would effectively put no pressure on the denoiser to learn the corrupted part of the target distribution, and focus instead on the frequency ranges holding the true signal.

We compare original and corrupted samples from MNIST, as well as samples from standard and *bpm*-SAGD-trained models in Figure 5. In line with our hypothesis, we observe frequency diffusion DPMs trained with an appropriate two-band SAGD to be able to discard the corrupting information and recover the original distribution after severe corruption. We further measure the FID and KID of the samples generated by the baseline and frequency DPMs against the original (uncorrupted) data samples in Table 2. We perform 8 ablation studies, considering noises at 0.1 non-overlapping

Table 2: Resulting FID and KID between standard diffusion and *bpm*-SAGD samplers trained on noise-corrupted data, with respect to samples from the true uncorrupted distribution (mean ± standard error across 3 seeds). We report eight ablation experiments across different non-overlapping corruption noise schemes.

| Dataset → | Baseline | | *bpm*-SAGD | |
|---|---|---|---|---|
| Corruption ↓ | FID (↓) | KID (↓) | FID (↓) | KID (↓) |
| $\epsilon_{[0.1,0.2]}$ | $3.2273_{\pm 8.50e-03}$ | $0.0114_{\pm 3.13e-05}$ | $2.7572_{\pm 3.56e-02}$ | $0.0095_{\pm 1.47e-04}$ |
| $\epsilon_{[0.2,0.3]}$ | $3.6601_{\pm 4.43e-03}$ | $0.0132_{\pm 1.67e-05}$ | $3.0416_{\pm 4.47e-02}$ | $0.0107_{\pm 1.79e-04}$ |
| $\epsilon_{[0.3,0.4]}$ | $3.4771_{\pm 4.79e-03}$ | $0.0125_{\pm 1.89e-05}$ | $2.9952_{\pm 3.35e-02}$ | $0.0106_{\pm 1.23e-04}$ |
| $\epsilon_{[0.4,0.5]}$ | $3.4281_{\pm 5.46e-03}$ | $0.0123_{\pm 1.98e-05}$ | $2.9218_{\pm 2.54e-02}$ | $0.0105_{\pm 8.79e-05}$ |
| $\epsilon_{[0.5,0.6]}$ | $3.3638_{\pm 6.31e-03}$ | $0.0121_{\pm 2.32e-05}$ | $2.8267_{\pm 2.81e-02}$ | $0.0102_{\pm 9.32e-05}$ |
| $\epsilon_{[0.6,0.7]}$ | $3.2444_{\pm 7.10e-03}$ | $0.0116_{\pm 2.55e-05}$ | $2.7026_{\pm 3.90e-02}$ | $0.0097_{\pm 1.28e-04}$ |
| $\epsilon_{[0.7,0.8]}$ | $3.0442_{\pm 6.32e-03}$ | $0.0109_{\pm 2.29e-05}$ | $2.5469_{\pm 6.39e-02}$ | $0.0091_{\pm 2.00e-04}$ |
| $\epsilon_{[0.8,0.9]}$ | $3.4660_{\pm 7.90e-03}$ | $0.0124_{\pm 2.96e-05}$ | $2.5138_{\pm 9.63e-02}$ | $0.0090_{\pm 3.07e-04}$ |

intervals in the $[0.1, .9]$ frequency range. We observe appropriately designed *bpm*-SAGD samplers to outperform standard diffusion training across all tested ranges. Interestingly, we observe better performance (lower FID) for data corruption in the high-frequency ranges, and reduced performance for data corruption in low-frequency ranges, confirming a marginally higher information content in the low frequencies for the MNIST dataset.

# 4 RELATED WORK

Beyond schedule tuning and architectural changes, several works have explicitly altered the forward corruption to shape inductive biases. Cold Diffusion replaces Gaussian noise with deterministic degradations (e.g., blur, masking), learning to invert arbitrary transforms without relying on stochastic noise (Bansal et al., 2023). Others introduce colored/ correlated Gaussian noise: for example, Huang et al. (2024) construct spatially correlated (blue/red) noise to emphasize selected frequency bands and show improved fidelity in image synthesis. More recently, frequency-domain guidance has been used to shape what diffusion models learn during sampling: FDG-Diff (Zhang et al., 2025) modulates feature spectra via a frequency-domain guidance module, and Frequency-Guided Diffusion (Gao et al., 2025) adjusts high-frequency components during text-driven image translation. Both operate in the frequency domain but retain an isotropic Gaussian forward process. Diffusion-like models based on reversing the heat equation similarly bias learning toward coarse-to-fine structure (Rissanen et al., 2023).

In contrast to these methods, SAGD retains a Gaussian forward model with frequency-diagonal covariance, providing a probabilistically consistent framework that is compatible with standard samplers (e.g., DDIM) and supported by a proof that, under full spectral support, the learned score converges to the true data score as $t \to 0$ (Sec. D). Closest to our setting, Voleti et al. (2022) formulate

diffusion with a non-isotropic Gaussian forward covariance $\Sigma$, and derive the corresponding denoising relations; their experiments, however, do not instantiate a spectrally structured operator and only show preliminary results on CIFAR-10. In contrast, we construct two noising operators while constraining $\Sigma$ to be diagonal in the Fourier basis (i.e., $\Sigma = \mathcal{F}^{-1}\mathrm{Diag}(|w(\mathbf{f})|^2)\mathcal{F}$), which (i) yields a simple, drop-in implementation via *per-frequency weighting* $\rightarrow$ IFFT, (ii) leaves the standard $\ell_2$ $\epsilon$-objective unchanged while enabling a closed-form score–$\epsilon$ conversion by multiplying with $\Sigma^{-1}$ (trivial in the spectral domain), and (iii) admits principled selective omission by zeroing prescribed bands. To our knowledge, this is the first forward–covariance manipulation method implemented with a frequency-diagonal Gaussian and validated through both spectral ablations and multi-dataset studies–while recovering standard isotropic Gaussian diffusion as a special case.

## 5 Discussion and Conclusion

**Summary.** In this work, we studied the potential to build inductive biases in the training and sampling of Diffusion Probabilistic Models (DPMs) by purposeful manipulation of the forward covariance in the noising process. We introduced spectrally anisotropic Gaussian diffusion (SAGD), an approach that guides DPMs via an anisotropic Gaussian forward operator with frequency-diagonal covariance. We compare SAGD to DPMs trained with standard Gaussian noise on generative visual tasks spanning datasets with significantly varying structures and scales. We show several key findings. First, we show that under full spectral support, the learned score converges to the true data score as $t \rightarrow 0$, while anisotropy deterministically reshapes the probability–flow path from noise to data. Second, we show how shaping the forward covariance serves as a strong inductive bias that steers diffusion samplers to better learn information at particular frequencies. Third, this property can be leveraged on both standard natural-image benchmarks (e.g., FFHQ, ImageNet-1k) and less conventional datasets (e.g., MNIST, DomainNet-Quickdraw), often yielding comparable or superior sampling quality to standard diffusion schedules, while remaining a minimal drop-in change to existing diffusion pipelines. Finally, SAGD enables selective omission, whereby by zeroing chosen bands the model ignores unwanted content and recovers clean signals in desired ranges.

**Future Work.** In our approach, we crafted two particular choices of SAGD forward-noise operators: a power-law weighting and a two-band mixture. In future work, several other alternatives may be considered, which can serve as more flexible tools to inject useful inductive biases for similar tasks. Moreover, the approach can be extended beyond constant schedules. For instance, time-varying spectral strategies ($\Sigma_{w_t}$) could shift focus from low-frequency (general shapes) to high-frequency (edges/textures) components over the sampling trajectory. Such methods could more closely align with human visual processing, which progressively sharpens details over time, offering a more natural sampling process. Additionally, other domains of noise manipulation, outside of spectral operations, may also present new opportunities for further steerability and improvements.

**Limitations and Considerations.** A current limitation lies in the complexity of relating spatial percepts to their frequency-domain representations. The perception of information in the frequency domain does not always translate straightforwardly to visual content, impeding the process of designing optimal ad hoc operators. In practice, empirical validation is still required to identify the best inductive biases for a given dataset. We believe it worthwhile for future work to develop analytical tools to guide operator design using data-specific considerations (e.g., spectral diagnostics and band-consistent metrics).

Finally, while our experiments focus on images, the approach applies to any domain with an intrinsic spectral basis in which the forward covariance can be specified and (approximately) diagonalized (e.g., 1D time series, 2D/3D grids and videos via Fourier/DCT, geospatial fields, and graph/mesh signals via Laplacian eigenbases). In contrast, domains lacking a coherent topology or spectral geometry (e.g., unordered sets or purely categorical tabular data) offer no natural basis for anisotropic forward covariances, making our construction less applicable.

**Conclusion** Overall, this work opens the door for more targeted and flexible diffusion generative modeling by building inductive biases through the manipulation of the forward nosing process. The ability to design noise schedules that align with specific data characteristics holds promise for advancing the state of the art in generative modeling.

**Reproducibility Statement.** We take several steps to facilitate the reproducibility of our results. The SAGD formulation and training objective are specified in Secs. 2.1–2.2 (forward step in Eq. equation 3, spectral operator in Eqs. equation 5–equation 6), with the score–$\epsilon$ relation, posterior, and probability–flow ODE derived in Appendix D. We provide minimal code to generate noise according to our two proposed SAGD operators (power-law and band-pass) in Appendix E. Datasets and preprocessing details appear in Appendices B and C; controlled corruptions and synthetic data procedures (power-law random fields) are documented alongside the code. All ablation settings (e.g., $\alpha$ grids, two-band ranges and weights, and seed counts) are enumerated in the Results tables/figures (e.g., **??**, Figure F.1). Finally, all evaluations follow standard FID/KID protocols using Inception-v3 (block 768); sample counts and metrics are reported per experiment.

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

## A    ANIMATED PARTICLE TRAJECTORIES (SUPPLEMENTARY VIDEOS)

We include three short animations of particle flows under the probability–flow ODE (deterministic DDIM; cf. Eq. equation 35). Each video shows $N$ particles transported from an isotropic Gaussian prior toward a fixed mixture-of-Gaussians target in $\mathbb{R}^2$.

**Files:**

- `particles_alpha-0p5git`: SAGD with power-law weighting $w_\alpha(r) = (r + \varepsilon)^\alpha$ (Eq. equation 10) and $\alpha = -0.5$ (low-frequency tilt).
- `particles_iso.gif`: isotropic baseline, $\alpha = 0$ (white-noise forward covariance).
- `particles_alpha0p5.gif`: SAGD with $\alpha = +0.5$ (high-frequency tilt).

All three runs share the same prior, target, and $\beta(t)$; only the forward covariance differs via $\Sigma_w = \mathcal{F}^{-1}\text{Diag}(|w_\alpha|^2)\mathcal{F}$. As time decreases $t : 1 \to 0$, there is a perceived spatial bias in the particle trajectory to the target. In all cases with full spectral support ($\Sigma_w \succ 0$), the endpoint score at $t \to 0$ is consistent with the true data score. In practice, the anisotropic path deviations favor the learning of particular aspects of the sampling distribution.

## B    DATASETS

For the experiments, we consider five datasets, namely: MNIST, CIFAR-10, Domainnet-Quickdraw, Wiki-Art and FFHQ; providing examples of widely different visual distributions, scales, and domain-specific statistics.

**MNIST:**    MNIST consists of $70,000$ grayscale images of handwritten digits (0-9) (Matthey et al., 2017). MNIST provides a simple test-bed to for the hypothesis in this work, as a well understood dataset with well structured, and visually coherent samples.

**CIFAR-10:**    CIFAR-10 contains $60,000$ color images distributed across 10 object categories (Krizhevsky et al., 2009). The dataset is highly diverse in terms of object appearance, backgrounds, and colors, with the wide-ranging visual variations across classes like animals, vehicles, and other common objects.

**DomainNet-Quickdraw:**    DomainNet-Quickdraw features $120,750$ sketch-style images, covering 345 object categories (Peng et al., 2019). These images, drawn in a minimalistic, abstract style, present a distribution that is drastically different from natural images, with sparse details and heavy visual simplifications.

**WikiArt:**    WikiArt consists of over $81,000$ images of artwork spanning a wide array of artistic styles, genres, and historical periods (Saleh & Elgammal, 2015). The dataset encompasses a rich and varied distribution of textures, color palettes, and compositions, making it a challenging benchmark for generative models, which must capture both the global structure and fine-grained stylistic variations that exist across different forms of visual art.

**FFHQ:**    The Flickr-Faces-HQ (FFHQ) dataset comprises 70,000 high-resolution, aligned face images (Karras et al., 2019), curated to increase diversity in age, ethnicity, pose, lighting, accessories (e.g., eyeglasses, hats), and backgrounds. The dataset offers a rich distribution of facial features, and it is a strong benchmark for testing generative models' ability to capture fine-grained identity-preserving details and global facial structure.

**ImageNet-1k:**    ImageNet-1k is a large-scale benchmark of over 1.2 million training images and 50,000 validation images, spanning 1,000 object categories drawn from everyday visual concepts such as animals, vehicles, tools, and scenes (Deng et al., 2009). The dataset consists of high-resolution high-variability natural images with complex backgrounds, diverse viewpoints, and significant intra-class variation in appearance, scale, and context, making it a challenging benchmark for generative models.

## C  Implementation details

### Data Preprocessing

We form a minimal preprocessing pipeline across datasets and vary only the target spatial resolution. We standardize pixel intensities to $[-1, 1]$ and resample each image to at three resolutions: $32 \times 32$ (MNIST, CIFAR-10, and DomainNet-Quickdraw); $64 \times 64$ (WikiArt), $254 \times 254$ (FFHQ, and Imagenet1k). MNIST is treated as single-channel (`channels = 1`), while all other datasets are RGB (`channels = 3`).

### Code base

For our small- and medium-scale diffusion experiments on MNIST, CIFAR-10, DomainNet-Quickdraw, WikiArt, and FFHQ, we build on the `diffusers` library from Hugging Face, using its standard DDPM-style training loop and replacing only the forward noise with our SAGD operators. All other components (optimizer, scheduler, and sampling code) follow the default configurations described in the main text. For large-scale ImageNet-1k experiments, we adapt the public codebase of Zheng et al. (2025), which implements latent DiT models in a DINOv2 feature space and has been shown to reach state-of-the-art performance in diffusion modeling at scale. In this setting, we again change only the forward noise generation to SAGD, leaving the architecture, optimizer, and training schedule unchanged.

### Denoiser architecture

For all 2D image experiments on MNIST, CIFAR-10, DomainNet-Quickdraw, WikiArt, and FFHQ, we use a standard U-Net denoiser with four resolution levels and channel widths $(32, 64, 128, 256)$ in the encoder (mirrored in the decoder), resulting in approximately 15.9M trainable parameters. This architecture is kept fixed across all SAGD and isotropic baselines. For ImageNet-1k, we use the DiT-based latent diffusion model from Zheng et al. (2025), operating in the DINOv2 latent space at 256×256 resolution with 196M parameters. We do not modify the DiT architecture or hyperparameters; SAGD is introduced solely as a drop-in replacement for the isotropic forward noise, demonstrating that our method is compatible with—and beneficial for—state-of-the-art large-scale diffusion setups.

### Power-Law Weighting implementation

**Discretization and batching.** In code, we construct the grid with $f_x(k) = \frac{k}{W} - \frac{1}{2}$ and $f_y(\ell) = \frac{\ell}{H} - \frac{1}{2}$ for $k \in \{0, \dots, W - 1\}$, $\ell \in \{0, \dots, H - 1\}$ (equivalently, `np.linspace(-0.5, 0.5, W)` and `H`). This follows the standard `fftfreq` convention, where frequencies are expressed in cycles per pixel on $[-\frac{1}{2}, \frac{1}{2}]$ and $\pm\frac{1}{2}$ correspond to Nyquist. If one instead parameterizes frequencies on $[-1, 1]$ via $f' = 2f$ (and hence $r'(f') = 2r(f)$), the same power-law shape can be recovered by defining $w'_\alpha(f') = (r'(f')/2 + \varepsilon)^\alpha$; any resulting global scale factor in $w$ (and thus in $\Sigma_w$) is absorbed into $\sigma_t^2$ or removed by the per-sample variance normalization used in our implementation, so the inductive bias depends only on the relative weighting across frequencies, not on the chosen numeric range. The weight $w_\alpha$ is broadcast across batch (and channels, if present). For convenience, one may multiply in the `fftshift`-centered domain and undo the shift before the inverse FFT; this is equivalent to multiplying in the unshifted domain since $w_\alpha$ is radial.

**Optional variance calibration.** To keep $\mathbb{E}\|\epsilon^{(\alpha)}\|_2^2$ roughly constant across $\alpha$, an energy-preserving scalar

$$
C_\alpha = \left( \frac{1}{HW} \sum_{u,v} |w_\alpha(f_{uv})|^2 \right)^{-\frac{1}{2}} \tag{15}
$$

can be applied in equation 11, i.e., $\mathbf{N}_{\text{freq}}^{(\alpha)} \leftarrow C_\alpha \mathbf{N}_{\text{freq}} \cdot w_\alpha$. (Our experiments omit this by default, matching the implementation in the main text.)

PRACTICAL CONSIDERATIONS.

We instantiate SAGD by sampling $\epsilon^{(w)}$ via FFTs, multiplying Fourier coefficients by a real, per-frequency weight $w(\mathbf{f})$, and inverting to the spatial domain (Eqs. equation 5–equation 6). The $\ell_2$ training objective equation 3 remains unchanged, and converting $\epsilon_\theta$ to a score only requires multiplying by $\Sigma_w^{-1}$ (see above). The computational overhead is non-existent at inference time, and negligible during training, as FFTs dominate and weights are broadcastable; for RGB, we apply $w$ per channel.

## D  NOISE PARAMETERIZATION, SCORES, AND FREQUENCY-BASED DYNAMICS

In this section, we wish to formalize the role of frequency diffusion in correctly learning the gradient of the log probability density of the data distribution at various noise levels (the score function). We model frequency-based corruption as an anisotropic Gaussian forward process, derive the score–$\epsilon$ relation for this general case, and prove that as $t \to 0$ the learned score converges to the true data score whenever all frequencies are represented (full-rank covariance). We also derive the reverse/posterior formulas and discuss how shaping the forward covariance changes the path to the score, shifting the information burden across frequencies. Finally, we formalize the selective-omission case when some bands are removed.

**Setup and notation.** Let $\alpha_t \in (0, 1)$ be the per-step scaling, $\bar{\alpha}_t = \prod_{s=1}^t \alpha_s$, and $\sigma_t^2 = 1 - \bar{\alpha}_t$. In standard DDPM, the forward marginal is

$$q(\mathbf{x}_t \mid \mathbf{x}_0) \;=\; \mathcal{N}\big(\sqrt{\bar{\alpha}_t}\,\mathbf{x}_0,\; \sigma_t^2\,\mathbf{I}\big), \tag{16}$$

and one trains an $\epsilon$-predictor $\epsilon_\theta(\mathbf{x}_t, t)$ by minimizing

$$\mathcal{L} \;=\; \mathbb{E}_{t, \mathbf{x}_0,\; \epsilon \sim \mathcal{N}(0, \mathbf{I})}\big[\; \|\epsilon - \epsilon_\theta(\mathbf{x}_t, t)\|_2^2 \;\big], \qquad \mathbf{x}_t = \sqrt{\bar{\alpha}_t}\,\mathbf{x}_0 + \sigma_t\,\epsilon. \tag{17}$$

The optimal predictor is $\epsilon_\theta^\star(\mathbf{x}_t, t) = \mathbb{E}[\epsilon \mid \mathbf{x}_t]$ and the true score relates to it via

$$\nabla_{\mathbf{x}_t} \log q_t(\mathbf{x}_t) \;=\; -\frac{1}{\sigma_t}\,\epsilon_\star(\mathbf{x}_t, t), \quad \text{where} \quad \epsilon_\star(\mathbf{x}_t, t) = \mathbb{E}[\epsilon \mid \mathbf{x}_t]. \tag{18}$$

### D.1  FREQUENCY-BASED FORWARD PROCESS AS ANISOTROPIC GAUSSIAN

Let $w(\mathbf{f}) > 0$ be a (time-independent) radial spectral weight and let $\mathcal{F}$ denote the discrete Fourier transform (unitary). The linear operator

$$\mathbf{T}_w \;:=\; \mathcal{F}^{-1} \circ \mathrm{Diag}\big(w(\mathbf{f})\big) \circ \mathcal{F} \tag{19}$$

maps spatial white noise to frequency-based noise. Writing $\xi \sim \mathcal{N}(0, \mathbf{I})$ and $\epsilon^{(w)} = \mathbf{T}_w \xi$, we have $\epsilon^{(w)} \sim \mathcal{N}(0, \Sigma_w)$ with

$$\Sigma_w \;=\; \mathbf{T}_w \mathbf{T}_w^\top \;=\; \mathcal{F}^{-1} \mathrm{Diag}\big(|w(\mathbf{f})|^2\big)\,\mathcal{F}, \tag{20}$$

i.e., $\Sigma_w$ is circulant and diagonalized by the Fourier basis, with eigenvalues given by the power spectrum $|w|^2$.[2]

Our forward process uses this shaped noise at each step:

$$\mathbf{x}_t \;=\; \sqrt{\alpha_t}\,\mathbf{x}_{t-1} \;+\; \sqrt{1 - \alpha_t}\,\epsilon_t^{(w)}, \qquad \epsilon_t^{(w)} \stackrel{\text{i.i.d.}}{\sim} \mathcal{N}(0, \Sigma_w). \tag{21}$$

A simple induction gives the marginal

$$q_w(\mathbf{x}_t \mid \mathbf{x}_0) \;=\; \mathcal{N}\big(\sqrt{\bar{\alpha}_t}\,\mathbf{x}_0,\; \sigma_t^2 \Sigma_w\big). \tag{22}$$

Hence, relative to equation 16, we have replaced the isotropic covariance by $\Sigma_w$, while $\bar{\alpha}_t$ and $\sigma_t^2$ remain unchanged.

**Support condition.**  If $w(\mathbf{f}) > 0$ for all $\mathbf{f}$, then $\Sigma_w \succ 0$ (full rank) and the forward kernels have full support in $\mathbb{R}^{HW}$. If $w$ vanishes on a band, $\Sigma_w$ is singular and the forward kernels are supported on a strict subspace (Section D.5). In practice, adding a small DC floor (e.g., $r(\mathbf{f}) \mapsto r(\mathbf{f}) + \varepsilon$ with $\varepsilon > 0$) ensures $w(0) > 0$ and thus $\Sigma_w \succ 0$.

---

[2] With the usual Hermitian pairing in the discrete Fourier basis, $\epsilon^{(w)}$ is real-valued.

## D.2 SCORE–$\epsilon$ RELATION UNDER ANISOTROPIC COVARIANCE

From equation 22,

$$\nabla_{\mathbf{x}_t} \log q_w(\mathbf{x}_t \mid \mathbf{x}_0) \;=\; -\frac{1}{\sigma_t^2}\,\Sigma_w^{-1}\big(\mathbf{x}_t - \sqrt{\bar{\alpha}_t}\,\mathbf{x}_0\big). \tag{23}$$

Taking the posterior expectation over $q_w(\mathbf{x}_0 \mid \mathbf{x}_t)$ and using $\mathbf{x}_t - \sqrt{\bar{\alpha}_t}\,\mathbf{x}_0 = \sigma_t\,\epsilon^{(w)}$, we obtain the marginal score

$$\nabla_{\mathbf{x}_t} \log q_{w,t}(\mathbf{x}_t) \;=\; -\frac{1}{\sigma_t}\,\Sigma_w^{-1}\,\underbrace{\mathbb{E}\big[\epsilon^{(w)} \,\big|\, \mathbf{x}_t\big]}_{:=\,\epsilon_\star^{(w)}(\mathbf{x}_t, t)}. \tag{24}$$

Training with the natural generalization of equation 17,

$$\mathcal{L}^{(w)} \;=\; \mathbb{E}_{t, \mathbf{x}_0,\, \epsilon^{(w)} \sim \mathcal{N}(0, \Sigma_w)}\big[\, \|\epsilon^{(w)} - \epsilon_\theta(\mathbf{x}_t, t)\|_2^2 \,\big], \qquad \mathbf{x}_t = \sqrt{\bar{\alpha}_t}\,\mathbf{x}_0 + \sigma_t\,\epsilon^{(w)}, \tag{25}$$

the optimal predictor is $\epsilon_\theta^\star(\mathbf{x}_t, t) = \mathbb{E}[\epsilon^{(w)} \mid \mathbf{x}_t]$. Therefore, a consistent score estimator is

$$s_\theta(\mathbf{x}_t, t) \;:=\; \nabla_{\mathbf{x}_t} \log q_{w,t}(\mathbf{x}_t) \;\approx\; -\frac{1}{\sigma_t}\,\Sigma_w^{-1}\,\epsilon_\theta(\mathbf{x}_t, t). \tag{26}$$

Equation equation 26 reduces to equation 18 when $\Sigma_w = \mathbf{I}$. Since the corruption covariance $\Sigma_w$ is fixed, the $\ell_2$ objective needs no reweighting—the optimal $\epsilon_\theta$ remains the conditional mean; $\Sigma_w^{-1}$ appears only when converting $\epsilon_\theta$ to the score via Eq. equation 26.

## D.3 TWEEDIE'S IDENTITY AND THE LIMIT $t \to 0$

Write the marginal as a (scaled) Gaussian smoothing of the data:

$$q_{w,t}(\mathbf{x}) \;=\; \int q(\mathbf{x}_0)\, \mathcal{N}\big(\mathbf{x}\,;\, \sqrt{\bar{\alpha}_t}\mathbf{x}_0,\, \sigma_t^2 \Sigma_w\big)\, d\mathbf{x}_0. \tag{27}$$

Let $\mathbf{z}_t := \mathbf{x}_t / \sqrt{\bar{\alpha}_t}$; then $\mathbf{z}_t = \mathbf{x}_0 + \tilde{\sigma}_t\,\epsilon^{(w)}$ with $\tilde{\sigma}_t^2 = \sigma_t^2 / \bar{\alpha}_t$. The anisotropic Tweedie identity gives

$$\mathbb{E}\big[\mathbf{x}_0 \,\big|\, \mathbf{z}_t\big] \;=\; \mathbf{z}_t \;+\; \tilde{\sigma}_t^2\,\Sigma_w\,\nabla_{\mathbf{z}_t} \log p_t(\mathbf{z}_t), \qquad p_t = \mathrm{law}(\mathbf{z}_t). \tag{28}$$

Equivalently, in the original variable,

$$\nabla_{\mathbf{x}_t} \log q_{w,t}(\mathbf{x}_t) \;=\; \frac{\sqrt{\bar{\alpha}_t}}{\sigma_t^2}\,\Sigma_w^{-1}\Big(\mathbb{E}[\mathbf{x}_0 \mid \mathbf{x}_t] - \frac{\mathbf{x}_t}{\sqrt{\bar{\alpha}_t}}\Big). \tag{29}$$

As $t \to 0$, $\bar{\alpha}_t \to 1$, $\sigma_t \to 0$, and $q_{w,t} \Rightarrow q$. If $\Sigma_w \succ 0$ and $q$ admits a locally positive $C^1$ density with $\nabla \log q \in L^1_{\mathrm{loc}}$, the anisotropic Gaussian mollifier is an approximate identity and

$$\lim_{t \to 0} \nabla_{\mathbf{x}_t} \log q_{w,t}(\mathbf{x}_t) \;=\; \nabla_{\mathbf{x}} \log q(\mathbf{x}) \quad \text{for a.e. } \mathbf{x}. \tag{30}$$

Intuitively, the anisotropic Gaussian kernel in equation 27 shrinks to a Dirac as $\sigma_t \to 0$ regardless of its orientation, so the smoothed score converges to the true data score. Combining equation 24–equation 30, the $\epsilon$-parameterization with frequency-based noise yields a correct score at $t = 0$, provided $\Sigma_w \succ 0$.

To visualize how frequency noising alters trajectories and score geometry through time, Fig. D.1 shows particle flows under the probability–flow ODE for isotropic noise (top), and *plw*-SAGD high ($\alpha$=0.1, middle), and low-frequency ($\alpha$=−0.1, bottom) tilt, while Fig. D.2 shows the corresponding score fields $\nabla_{\mathbf{x}} \log p_t(\mathbf{x})$ at five equally spaced times. Frequency noising changes the path deterministically by reweighting modes through $\Sigma_w$, while preserving the $t \to 0$ endpoint score under full support (see Sec. D).

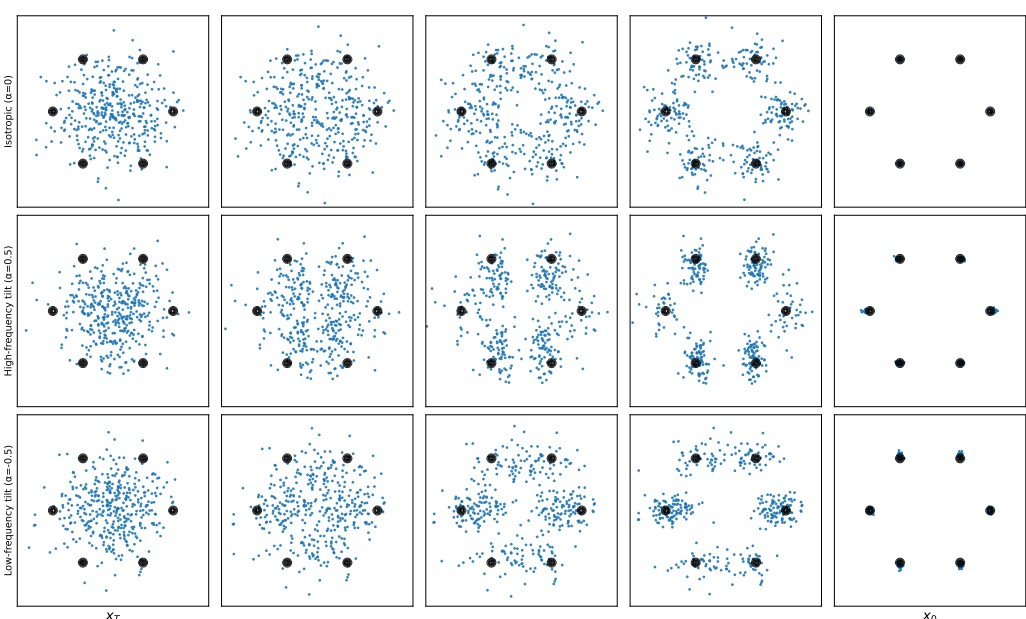

Figure D.1: Particle trajectories under the probability–flow ODE from a Gaussian prior to a mixture-of-Gaussians target (black contours), visualized at five equally spaced times (left to right). Rows: (top) isotropic noise ($\alpha=0$), (middle) high-frequency tilt ($\alpha=0.1$), (bottom) low-frequency tilt ($\alpha=-0.1$). *plw*-SAGD alters the path by reweighting modes via $\Sigma_w$ while keeping the endpoint consistent under full support (cf. Sec. D).

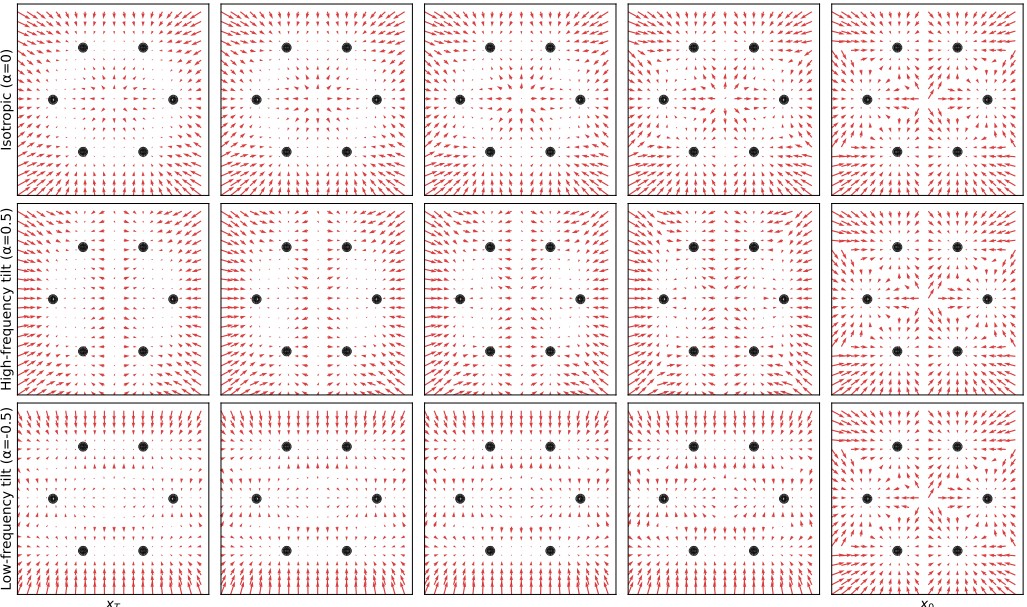

Figure D.2: Evolving score fields $\nabla_{\mathbf{x}} \log p_t(\mathbf{x})$ for the same three settings as Fig. D.1. Arrows indicate the instantaneous score on a grid; black contours show the target density. *plw*-SAGD stretches/compresses the field along principal modes, biasing the trajectory toward frequencies emphasized by $\Sigma_w$.

### D.4 REVERSE/POSTERIOR WITH FREQUENCY-BASED NOISE

Since all covariances are proportional to the same $\Sigma_w$, linear-Gaussian posteriors retain the standard scalar coefficients while the covariances inherit $\Sigma_w$ as a factor. In particular,

$$q_w(\mathbf{x}_{t-1} \mid \mathbf{x}_t, \mathbf{x}_0) \;=\; \mathcal{N}\!\Big(\tilde{\mu}_t(\mathbf{x}_t, \mathbf{x}_0),\, \tilde{\beta}_t\, \Sigma_w\Big), \tag{31}$$

with

$$\tilde{\beta}_t = \frac{1 - \bar{\alpha}_{t-1}}{1 - \bar{\alpha}_t}\, (1 - \alpha_t), \tag{32}$$

$$\tilde{\mu}_t(\mathbf{x}_t, \mathbf{x}_0) = \frac{1}{\sqrt{\alpha_t}}\left(\mathbf{x}_t \;-\; \frac{1 - \alpha_t}{1 - \bar{\alpha}_t}(\mathbf{x}_t - \sqrt{\bar{\alpha}_t}\, \mathbf{x}_0)\right). \tag{33}$$

Replacing $\mathbf{x}_0$ by $\hat{\mathbf{x}}_0$ in equation 33 yields the usual mean update. For the $\epsilon$-parameterization we recover an estimate of $\mathbf{x}_0$ via

$$\hat{\mathbf{x}}_0(\mathbf{x}_t, t) \;=\; \frac{1}{\sqrt{\bar{\alpha}_t}}\,(\mathbf{x}_t - \sigma_t\, \epsilon_\theta(\mathbf{x}_t, t)). \tag{34}$$

**Stochastic sampling:** if one samples stochastically (e.g., DDPM), the injected noise should be drawn as $\eta_t^{(w)} \sim \mathcal{N}(0, \Sigma_w)$ (not $\mathcal{N}(0, \mathbf{I})$) for consistency with the forward process.

**Probability-flow ODE:** if one uses the deterministic sampler (e.g., DDIM), no step noise is injected. In continuous time, the associated probability-flow ODE with frequency-based forward noise reads

$$\frac{d\mathbf{x}}{dt} \;=\; -\tfrac{1}{2}\,\beta(t)\,\mathbf{x} \;-\; \tfrac{1}{2}\,\beta(t)\,\Sigma_w\,\nabla_{\mathbf{x}} \log p_t(\mathbf{x}), \tag{35}$$

which reduces to the standard probability–flow ODE when $\Sigma_w = \mathbf{I}$. In practice with the $\epsilon$-parameterization, one uses $\hat{\mathbf{x}}_0$ from equation 34 in the standard DDIM deterministic update; no extra noise term appears.

### D.5 SELECTIVE OMISSION AND RANK-DEFICIENT $\Sigma_w$

If $w$ vanishes on a measurable band, then $\Sigma_w \succeq 0$ is singular. The forward kernels in equation 22 are supported on an affine subspace determined by $\mathrm{range}(\Sigma_w)$, and the smoothed marginals $q_{w,t}$ are not strictly positive in $\mathbb{R}^{HW}$. The score $\nabla \log q_{w,t}$ exists only on that subspace and is undefined along the null space. Training with equation 25 then recovers the projected score, i.e., the model learns to ignore the omitted bands by construction (this is the mechanism exploited in our corruption-recovery experiments).

### D.6 PATH TO THE SCORE EFFECTS OF SAGD

Even though the $t \to 0$ limit recovers the true data score under $\Sigma_w \succ 0$, the evolution of the score with $t$ changes substantially:

1. **Geometry of the score.** From equation 24, the conversion from $\epsilon$-prediction to score multiplies by $\Sigma_w^{-1}$. In the Fourier basis (where $\Sigma_w$ is diagonal), modes with larger variance (large $|w|^2$) are downweighted in the score, while low-variance modes are amplified. Thus, shaping the forward spectrum changes the relative gradient magnitudes across frequencies during training and sampling.

2. **Signal-to-noise during supervision.** The target $\epsilon^{(w)}$ has covariance $\Sigma_w$, so its per-mode variance follows $|w|^2$. The $\ell_2$ loss in equation 25 therefore exposes the model to larger target amplitudes (and larger gradients) in bands where $|w|$ is large, shifting the inductive bias toward fitting those modes sooner/more accurately.

3. **Reverse dynamics.** The reverse posterior covariance in equation 31 is $\tilde{\beta}_t \Sigma_w$, so the stochasticity injected at each reverse step is anisotropic. This changes the trajectory taken from $t$ down to 0, biasing the generation process to consolidate structure along directions favored by $\Sigma_w$. Under DDIM, no step noise is instead injected, so the anisotropic stochastic

effect disappears. However, the drift in the probability-flow ODE equation 35 still carries $\Sigma_w$ through the term $-\beta(t)\,\Sigma_w\,\nabla_{\mathbf{x}}\log p_t(\mathbf{x})$, inducing trajectories to be frequency-biased deterministically. In this context, modes emphasized by $\Sigma_w$ contribute more strongly to the drift, reshaping the path from $t{=}T$ to $t{=}0$ even without randomness.

Collectively, these effects explain why different datasets benefit from different $w$: the endpoint score is consistent (under full rank), but the path—and thus the optimization landscape and sample trajectories—is reshaped by frequency weighting.

**Time-varying weights.** If one uses a schedule $w_t(\mathbf{f})$, the $t$-step marginal covariance becomes a scalar-weighted sum of commuting matrices:

$$\mathrm{Cov}(\mathbf{x}_t \mid \mathbf{x}_0) \;=\; \sum_{s=1}^{t}\left(\beta_s \prod_{k=s+1}^{t}\alpha_k\right)\Sigma_{w_s}, \qquad \beta_s := 1-\alpha_s. \tag{36}$$

When all $\Sigma_{w_s}$ are diagonal in the Fourier basis (true for any per-frequency diagonal weight, not necessarily radial), the analysis carries through modewise with eigenvalues replaced by the corresponding positive weighted sums $\sum_s w_s\,|w_s(\mathbf{f})|^2$ (which form a convex combination after normalization by $\sigma_t^2 = \sum_s w_s$).

## E  REFERENCE CODE FOR SAGD NOISE GENERATORS

Below we provide minimal, implementation-oriented pseudo-code for the two SAGD operators used in this paper: (i) a *power-law* (radial) weighting and (ii) a *band-pass* mask (building block for the two-band mixture). Both follow the same template:

$$\boldsymbol{\epsilon}\sim\mathcal{N}(0,\mathbf{I}) \;\xrightarrow{\;\mathcal{F}\;}\; \mathbf{N}_{\mathrm{freq}} \;\xrightarrow{\;w(\mathbf{f})\;}\; \mathbf{N}_{\mathrm{freq}}^{(w)} \;\xrightarrow{\;\mathcal{F}^{-1}\;}\; \boldsymbol{\epsilon}^{(w)}.$$

We normalize the spatial noise to unit variance so that the overall scale still comes from the schedule via $\sigma_t$. Multiplying by a *real*, per-frequency weight preserves Hermitian symmetry and thus yields a real inverse transform.

**Power-law weighting.** This implements a *plw*-SAGD with $w_\alpha(\mathbf{f}) = (r(\mathbf{f})+\varepsilon)^\alpha,\, r(\mathbf{f}) = \sqrt{f_x^2 + f_y^2}$ and a small DC floor $\varepsilon > 0$.

```python
import numpy as np

def sagd_powerlaw_noise(shape, alpha, eps=1e-10):
    # shape: (B, C, H, W)
    B, C, H, W = shape
    # 1) white noise
    eps_white = np.random.randn(B, C, H, W)
    # 2) FFT over spatial axes
    F = np.fft.fftn(eps_white, axes=(-2, -1))
    # 3) radial grid in normalized frequency (cycles/pixel)
    fy = np.fft.fftfreq(H)[:, None]      # shape Hx1
    fx = np.fft.fftfreq(W)[None, :]      # shape 1xW
    r  = np.sqrt(fx**2 + fy**2)          # shape HxW
    # 4) power-law weight
    w  = (r + eps)**alpha                # shape HxW (real)
    # 5) apply weight and invert
    Fw = F * w[None, None, ...]
    eps_w = np.fft.ifftn(Fw, axes=(-2, -1)).real
    # 6) unit-variance normalization (per-sample)
    std = eps_w.std(axis=(-2, -1), keepdims=True) + 1e-8
    return eps_w / std
```

**Band-pass mask (single band $[a, b]$).** This implements a *bpm*-SAGD with $w_{a,b}(\mathbf{f}) = \mathbf{1}\{a \leq r(\mathbf{f}) \leq b\}$. Cutoffs $a, b$ are given in frequency units of cycles/pixel (e.g., $a$=0.1, $b$=0.3); we can also express them as fractions of the Nyquist radius.

```python
import numpy as np

def sagd_bandpass_noise(shape, a, b):
    # shape: (B, C, H, W); choose 0 <= a <= b <= ~sqrt(2)/2 (cycles/pixel)
    B, C, H, W = shape
    # 1) white noise
    eps_white = np.random.randn(B, C, H, W)
    # 2) FFT over spatial axes
    F = np.fft.fftn(eps_white, axes=(-2, -1))
    # 3) radial grid in normalized frequency (cycles/pixel)
    fy = np.fft.fftfreq(H)[:, None]     # shape Hx1
    fx = np.fft.fftfreq(W)[None, :]     # shape 1xW
    r  = np.sqrt(fx**2 + fy**2)         # shape HxW
    # 4) band-pass mask (keep a <= r <= b)
    M  = ((r >= a) & (r <= b)).astype(float)  # shape HxW
    # 5) apply mask and invert
    Fw = F * M[None, None, ...]
    eps_w = np.fft.ifftn(Fw, axes=(-2, -1)).real
    # 6) unit-variance normalization (per-sample)
    std = eps_w.std(axis=(-2, -1), keepdims=True) + 1e-8
    return eps_w / std
```

**Two-band mixture.** We combine two band-pass noises with nonnegative coefficients $\gamma_l, \gamma_h$ (typically $\gamma_l + \gamma_h$=1):

$$\boldsymbol{\epsilon}^{(\mathrm{w})} = \gamma_l \, \texttt{sagd\_bandpass\_noise}(\cdot, a_l, b_l) + \gamma_h \, \texttt{sagd\_bandpass\_noise}(\cdot, a_h, b_h).$$

**SAGD Forward step.** Replace the isotropic noise in the DDPM forward step with either generator above:

$$\mathbf{x}_t = \sqrt{\alpha_t} \, \mathbf{x}_{t-1} + \sqrt{1 - \alpha_t} \, \boldsymbol{\epsilon}^{(w)}.$$

The training loss and DDIM update remain unchanged; when converting $\epsilon_\theta$ to a score, multiply by $\Sigma_w^{-1}$ (diagonal in the Fourier basis).

# F  ADDITIONAL RESULTS

## F.1  SAGD IN NATURAL DATASETS - FULL RESULTS

We present in Table F.1 and Table F.2 the full set of results from running pwd-SAGD on all datasets and values of $\alpha$ considered. In Figure F.1, we can better discern the learning performance over different $\alpha$ settings, where we observe interesting monotonically decreasing performance (increasing FID) for MNIST and DomainNet, showcasing improved learning for negative values of $\alpha$, and suggesting that semantically informative content (strokes/contours) for these datasets may lie in the lower frequency ranges. By contrast, CIFAR-10, Wiki-Art, and FFHQ exhibit a much flatter dependence with shallow optima near $\alpha \approx 0$ (within ±0.02), consistent with their broader, mixed spectra. In Figure F.1, we can better discern the learning performance over different $\alpha$ settings, where we observe interesting monotonically decreasing performance (increasing FID) for MNIST and DomainNet, showcasing improved learning for negative values of $\alpha$, and suggesting that semantically informative content (strokes/contours) for these datasets may lie in the lower frequency ranges. By contrast, CIFAR-10, Wiki-Art, and FFHQ exhibit a much flatter dependence with shallow optima near $\alpha \approx 0$ (within ±0.02), consistent with their broader, mixed spectra.

Table F.1: Results for FID across different $\alpha$ (frequency slope) settings (mean ± standard error across seeds). The setting for $\alpha = 0$ corresponds to standard DPM training (baseline).

| Dataset → | MNIST | CIFAR-10 | Domainnet-Quickdraw | Wiki-Art | FFHQ | ImageNet1k |
|---|---|---|---|---|---|---|
| baseline | $0.42_{\pm 8.52e-03}$ | $0.75_{\pm 0.01}$ | $0.60_{\pm 0.05}$ | $1.06_{\pm 0.03}$ | $1.11_{\pm 0.01}$ | $8.6819_{\pm 0.0739}$ |
| $\alpha = -0.080$ | $0.31_{\pm 0.03}$ | $0.98_{\pm 0.03}$ | $0.55_{\pm 0.04}$ | $1.39_{\pm 0.10}$ | $1.48_{\pm 0.02}$ | - |
| $\alpha = -0.060$ | $0.28_{\pm 0.02}$ | $0.94_{\pm 0.02}$ | $0.52_{\pm 0.05}$ | $1.35_{\pm 0.08}$ | $1.74_{\pm 0.10}$ | $8.1098_{\pm 0.0229}$ |
| $\alpha = -0.040$ | $0.31_{\pm 7.76e-03}$ | $0.86_{\pm 0.02}$ | $0.49_{\pm 0.02}$ | $1.25_{\pm 0.07}$ | $1.76_{\pm 0.08}$ | $7.5534_{\pm 0.0556}$ |
| $\alpha = -0.020$ | $0.37_{\pm 6.36e-03}$ | $0.76_{\pm 0.01}$ | $0.52_{\pm 0.03}$ | $1.14_{\pm 0.05}$ | $1.68_{\pm 0.19}$ | $7.6419_{\pm 0.0581}$ |
| $\alpha = -0.010$ | $0.37_{\pm 0.02}$ | $0.75_{\pm 0.01}$ | $0.54_{\pm 0.04}$ | $1.09_{\pm 0.04}$ | $1.48_{\pm 0.12}$ | $8.0400_{\pm 0.0236}$ |
| $\alpha = -0.001$ | $0.40_{\pm 0.02}$ | $0.76_{\pm 0.01}$ | $0.56_{\pm 0.04}$ | $1.02_{\pm 5.66e-03}$ | $1.04_{\pm 5.17e-03}$ | $8.5288_{\pm 0.0112}$ |
| $\alpha = 0.001$ | $0.39_{\pm 0.02}$ | $0.76_{\pm 0.02}$ | $0.58_{\pm 0.03}$ | $1.02_{\pm 6.63e-03}$ | $1.45_{\pm 0.19}$ | - |
| $\alpha = 0.010$ | $0.43_{\pm 0.02}$ | $0.80_{\pm 0.02}$ | $0.66_{\pm 0.02}$ | $1.20_{\pm 0.07}$ | $2.06_{\pm 0.06}$ | $9.3867_{\pm 0.0348}$ |
| $\alpha = 0.020$ | $0.47_{\pm 0.02}$ | $0.85_{\pm 0.01}$ | $0.72_{\pm 0.03}$ | $1.40_{\pm 0.05}$ | $2.34_{\pm 0.04}$ | - |
| $\alpha = 0.040$ | $0.56_{\pm 0.02}$ | $0.95_{\pm 0.02}$ | $0.90_{\pm 0.04}$ | $1.47_{\pm 0.01}$ | $2.81_{\pm 0.03}$ | - |
| $\alpha = 0.060$ | $0.65_{\pm 0.04}$ | $1.06_{\pm 0.04}$ | $1.22_{\pm 0.05}$ | $1.56_{\pm 0.04}$ | $2.97_{\pm 0.03}$ | - |
| $\alpha = 0.080$ | $0.78_{\pm 0.04}$ | $1.15_{\pm 0.05}$ | $1.52_{\pm 0.05}$ | $1.58_{\pm 0.04}$ | $3.10_{\pm 9.95e-03}$ | - |

Table F.2: Results for KID across different $\alpha$ (frequency slope) settings (mean ± standard error across seeds). The setting for $\alpha = 0$ corresponds to standard DPM training (baseline).

| Dataset → | MNIST | CIFAR-10 | Domainnet-Quickdraw | Wiki-Art | FFHQ |
|---|---|---|---|---|---|
| baseline | $9.16\text{e-}04_{\pm 6.44e-05}$ | $2.11\text{e-}04_{\pm 1.15e-05}$ | $7.15\text{e-}04_{\pm 1.06e-04}$ | $8.08\text{e-}04_{\pm 1.28e-04}$ | $1.42\text{e-}03_{\pm 8.12e-05}$ |
| $\alpha = -0.080$ | $5.63\text{e-}04_{\pm 7.19e-05}$ | $8.64\text{e-}04_{\pm 6.72e-05}$ | $6.54\text{e-}04_{\pm 5.45e-05}$ | $2.03\text{e-}03_{\pm 3.32e-04}$ | $2.79\text{e-}03_{\pm 1.01e-04}$ |
| $\alpha = -0.060$ | $4.53\text{e-}04_{\pm 8.19e-05}$ | $7.42\text{e-}04_{\pm 5.04e-05}$ | $5.28\text{e-}04_{\pm 8.75e-05}$ | $1.97\text{e-}03_{\pm 3.32e-04}$ | $3.81\text{e-}03_{\pm 3.43e-04}$ |
| $\alpha = -0.040$ | $5.29\text{e-}04_{\pm 5.44e-05}$ | $4.71\text{e-}04_{\pm 5.95e-05}$ | $4.73\text{e-}04_{\pm 1.54e-05}$ | $1.69\text{e-}03_{\pm 2.94e-04}$ | $3.75\text{e-}03_{\pm 3.14e-04}$ |
| $\alpha = -0.020$ | $7.87\text{e-}04_{\pm 2.19e-05}$ | $2.25\text{e-}04_{\pm 1.42e-05}$ | $5.47\text{e-}04_{\pm 4.14e-05}$ | $1.32\text{e-}03_{\pm 1.50e-04}$ | $3.44\text{e-}03_{\pm 6.55e-04}$ |
| $\alpha = -0.010$ | $7.62\text{e-}04_{\pm 8.03e-05}$ | $1.90\text{e-}04_{\pm 2.01e-05}$ | $5.87\text{e-}04_{\pm 7.17e-05}$ | $9.64\text{e-}04_{\pm 1.16e-04}$ | $2.82\text{e-}03_{\pm 3.94e-04}$ |
| $\alpha = -0.001$ | $8.32\text{e-}04_{\pm 6.08e-05}$ | $2.29\text{e-}04_{\pm 3.56e-05}$ | $6.59\text{e-}04_{\pm 8.39e-05}$ | $6.87\text{e-}04_{\pm 7.94e-05}$ | $1.23\text{e-}03_{\pm 5.66e-05}$ |
| $\alpha = 0.001$ | $8.00\text{e-}04_{\pm 6.49e-05}$ | $2.49\text{e-}04_{\pm 3.57e-05}$ | $7.02\text{e-}04_{\pm 6.25e-05}$ | $7.21\text{e-}04_{\pm 7.45e-05}$ | $2.48\text{e-}03_{\pm 6.59e-04}$ |
| $\alpha = 0.010$ | $9.49\text{e-}04_{\pm 8.17e-05}$ | $3.28\text{e-}04_{\pm 2.90e-05}$ | $8.99\text{e-}04_{\pm 5.53e-05}$ | $1.36\text{e-}03_{\pm 2.58e-04}$ | $4.74\text{e-}03_{\pm 1.00e-04}$ |
| $\alpha = 0.020$ | $1.09\text{e-}03_{\pm 6.57e-05}$ | $5.05\text{e-}04_{\pm 2.52e-05}$ | $1.06\text{e-}03_{\pm 7.36e-05}$ | $2.18\text{e-}03_{\pm 2.05e-04}$ | $5.80\text{e-}03_{\pm 1.26e-04}$ |
| $\alpha = 0.040$ | $1.39\text{e-}03_{\pm 7.64e-05}$ | $8.03\text{e-}04_{\pm 5.90e-05}$ | $1.51\text{e-}03_{\pm 1.30e-04}$ | $2.45\text{e-}03_{\pm 1.26e-04}$ | $7.65\text{e-}03_{\pm 8.05e-05}$ |
| $\alpha = 0.060$ | $1.72\text{e-}03_{\pm 1.53e-04}$ | $1.12\text{e-}03_{\pm 8.61e-05}$ | $2.37\text{e-}03_{\pm 1.47e-04}$ | $2.83\text{e-}03_{\pm 2.40e-04}$ | $8.37\text{e-}03_{\pm 1.59e-04}$ |
| $\alpha = 0.080$ | $2.18\text{e-}03_{\pm 1.59e-04}$ | $1.37\text{e-}03_{\pm 1.09e-04}$ | $3.30\text{e-}03_{\pm 1.68e-04}$ | $2.94\text{e-}03_{\pm 1.84e-04}$ | $8.87\text{e-}03_{\pm 7.44e-05}$ |

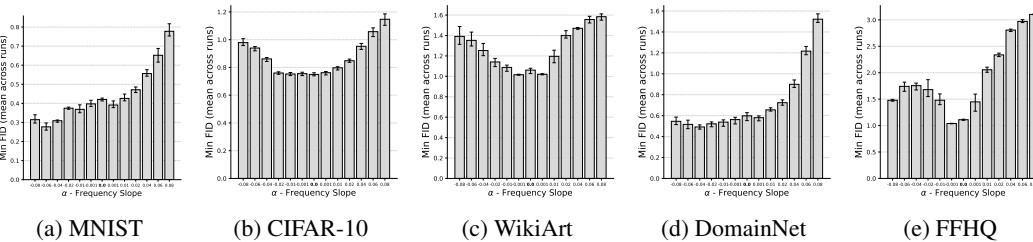

|     |     |     |     |     |
|---|---|---|---|---|
| (a) MNIST | (b) CIFAR-10 | (c) WikiArt | (d) DomainNet | (e) FFHQ |

Figure F.1: Minimum FID vs. frequency slope ($\alpha$). Bars show mean FID with inter-quartile error bars across runs across three seeds.

