# OpenReview forum: "Learning What Matters: Steering Diffusion via Spectrally Anisotropic Forward Noise"
_ICLR.cc/2026/Conference — Submitted to ICLR 2026_

### Official Review · Reviewer_SWmF · 2025-10-16

**Soundness:** 3
**Presentation:** 3
**Contribution:** 2
**Rating:** 4
**Confidence:** 4

**Summary:**

This paper explores introducing anisotropic noise to the forward process of diffusion models, as a method of controlling the inductive biases present during training to better match that of the training data. This is done by reweighing the added noise in the Fourier domain, either by a radial power-law reweighing of frequencies or by a band-pass filter. This method is then evaluated experimentally on various image datasets.

Although the questions this paper set out to answer are interesting, the proposed method seems ad-hoc and makes choices that are not well justified. The experimental evidence is also not strong, and it is unclear that this method is effective on larger image datasets.

**Strengths:**

Shaping the added noise in diffusion models to better match the data distribution is important to further improve their performance, and this paper proposes a new method of doing so. There are also extensive experimental evaluations on many image datasets.

**Weaknesses:**

1. There is not much motivation on why the Fourier basis is chosen, or for the design choices of band-pass/power-law-weighted filters. I am not convinced why adding spectrally anistropic noise is better than white noise for image datasets.
2. The theoretical results only show that the true data distribution can be recovered, but it does not provide any reason for why adding anisotropic noise would be helpful in training and sampling of diffusion models. It would be more compelling if there are results showing what is the optimal noise to add for any given data statistics (e.g. average power spectral density).
3. The results (Fig. 4) only show significant improvement for MNIST, but the limited improvements on larger datasets cast doubt on the practicality of this method on larger-scale and more diverse datasets.

**Questions:**

1. Instead of the Fourier basis and hand-designed filters, is it possible to learn a basis from training data?
2. Is this equivalent to applying a linear transformation on data, then performing standard denoising using white noise in the transformed space?
3. If the power-spectral density of natural images follow a power law, then adding white Gaussian noise will first corrupt high-frequency components before low-frequency ones. In this case, how would using spectrally anistropic noise be different from using a different schedule for white noise?
4. For FID evaluations, why use the 768-dim layer instead of the standard 2048-dim layer?
5. Is it possible to implement this method on a state-of-the-art baseline (e.g. EDM) and see if generation quality is improved?

---

> ### Author Response · Authors · 2025-11-21
>
> We thank the reviewer for engaging with the paper and for highlighting both the importance of shaping noise in diffusion models and the breadth of our experimental evaluation.
>
> Below we address your points in turn, with emphasis on (i) the motivation for Fourier-domain design, (ii) what SAGD adds beyond standard white-noise schedules, and (iii) evidence on large-scale, “real-world” image datasets and state-of-the-art baselines.

---

> ### Author Response · Authors · 2025-11-21
>
> ### 1) Motivation for the Fourier basis and the choice of filters
>
> > *“There is not much motivation on why the Fourier basis is chosen, or for the design choices of band-pass/power-law-weighted filters. I am not convinced why adding spectrally anisotropic noise is better than white noise for image datasets.”*
>
> We chose the Fourier basis for three main reasons, which we have now made more explicit in the manuscript:
>
> 1. **Stationary covariances diagonalize in the Fourier basis.**
>    Any stationary Gaussian field has a covariance that is diagonal in the Fourier basis, with eigenvalues given by its power spectrum. This makes the Fourier domain a natural place to specify structured covariances with clear semantics.
>
> 2. **Direct control of spectral content and selective omission.**
>    Rather than hand-crafted filters, the power-law family proposed and band-pass masks are **low-dimensional parameterizations of the eigenvalues** of the forward covariance:
>    - The power-law family $(r(f)+\varepsilon)^\alpha$ spans a continuum from low–frequency-tilted to high–frequency-tilted spectra, with $\alpha=0$ recovering the standard isotropic Gaussian.
>    - The two-band mixture (bpm-SAGD) allows us to **zero out** or suppress specific bands, enabling *selective omission* (e.g., ignore structured artifacts in known frequency ranges).
>
>    In pixel space, these operations correspond to long-range, dense covariances that are much less transparent to design and analyze, while the frequency formulation gives us a very simple handle to control these.
>
> 3. **Simplicity and empirical impact**
>    As for many good "tricks" we have found it to be both very simple to implement in any codebase (≈40 lines of code to replace standard Gaussian noise in the forward process to SAGD noise), and strong in practice, with measurable gains on most tested datasets.
>
> We do not wish to claim that spectrally anisotropic noise is *universally* better than white noise for every dataset; rather, SAGD provides a **principled, interpretable knob** to align the forward process with known or desired spectral properties of the data, sometimes yielding substantial improvements and, at worst, recovering the standard isotropic baseline.
>
> We have clarified these design motivations and the role of the Fourier basis in the revised *Methods* section.

---

> ### Author Response · Authors · 2025-11-21
>
> ### 2) Theory, “optimal noise”, and learned bases
>
> > *“The theoretical results only show that the true data distribution can be recovered, but it does not provide any reason for why adding anisotropic noise would be helpful… It would be more compelling if there are results showing what is the optimal noise to add for any given data statistics (e.g. average power spectral density).”*
> > *“Instead of the Fourier basis and hand-designed filters, is it possible to learn a basis from training data?”*
>
> Thank you for this comment. Indeed, fully characterizing the *optimal* forward covariance for arbitrary data statistics is a very interesting avenue.
>
> We did explore data-driven heuristics (similar to what the reviewer suggests), including:
>
> - Designing $w(f)$ directly from empirical power spectra (e.g., “flattening” the spectrum or up-/down-weighting dominant bands),
> - Using radially averaged spectra as guidance.
>
> In our experiments, these simple mappings did not yield a robust rule that consistently outperforms the simple scalar $\alpha$ sweeps, suggesting that the relationship between average PSD and optimal $w(f)$ is more subtle. Ultimately, we designed the best performing SAGD variants so that a simple 1-dimensional sweep of alpha is sufficient to tune the model.
>
> In practice:
>
> - We view SAGD as a **minimal, tractable family** of structured covariances that:
>   1. Strictly generalizes isotropic Gaussian noise ($\alpha=0$),
>   2. Let us smoothly tilt spectral emphasis via a single scalar $\alpha$, and
>   3. Supports hard inclusion/exclusion of bands (bpm-SAGD).
>
> - We empirically show that, within this family, different choices of $\alpha$ and band masks **trade off performance** in ways that correlate with the spectral structure of the data:
>   e.g., low-frequency tilts help on datasets where information is concentrated in coarse structure, and selective omission helps when corruptions live in known bands.
>
>
> We have updated the *Discussion* section to better clarify these aspects.

---

> ### Author Response · Authors · 2025-11-21
>
> ### 3) Limited improvements on larger datasets and state-of-the-art baselines
>
> > *“The results (Fig. 4) only show significant improvement for MNIST, but the limited improvements on larger datasets cast doubt on the practicality of this method on larger-scale and more diverse datasets.”*
> > *“Is it possible to implement this method on a state-of-the-art baseline (e.g. EDM) and see if generation quality is improved?”*
>
> Thank you for raising this point, and indeed it is! We have extended our results with **additional experiments on ImageNet-1k** at 256×256 using a recent **state-of-the-art DiT-based latent diffusion implementation (RAE/DiT)**, achieving state-of-the-art results at scale [1]:
>
> - **Codebase / model:** public RAE/DiT implementation (DINOv2 latent space), $\approx196M$ parameters.
> - **Dataset:** ImageNet-1k (1.28M images), resolution 256×256.
> - **Training setup:** global batch 1024 (micro-batch 128), AdamW (lr $2\cdot10^{-4}$, betas $(0.9, 0.95)$), schedule and architecture exactly as in the original RAE paper.
>
> **Importantly,** we only replace isotropic forward noise with SAGD and sweep a small 1D grid of $\alpha \in \{\[-0.06,-0.04,-0.02,-0.01,0.0,+0.01\]\}$; all other hyperparameters are kept fixed.
>
> We obtain (mean ± standard error over seeds):
>
> | $\alpha$   | FID ↓            |
> |-----------:|------------------|
> | -0.0600    | 8.11 ± 0.02      |
> | **-0.0400**    | **7.55 ± 0.06**  |
> | -0.0200    | 7.64 ± 0.06      |
> | -0.0100    | 8.04 ± 0.02      |
> | 0.0000     | 8.68 ± 0.07      |
> | 0.0100     | 9.39 ± 0.03      |
>
> This shows that:
>
> - **On a large-scale, natural-image dataset with a strong modern baseline**, SAGD improves FID from **8.68 → 7.55** at 256×256, without any architectural changes or tuning beyond a 1D $\alpha$ sweep.
> - The optimum is at a **non-zero** $\alpha=-0.04$; performance degrades for more negative values and for positive values, indicating a low-frequency-tilted covariance is beneficial at scale.
>
> We have integrated these ImageNet results into the main *Results* section and added implementation details in the appendices.
>
> #### **References**
> [1] Zheng, B., Ma, N., Tong, S. and Xie, S., 2025. Diffusion Transformers with Representation Autoencoders. arXiv preprint arXiv:2510.11690.

---

> ### Author Response · Authors · 2025-11-21
>
> ### 4) Equivalence to a linear transformation
>
> > *“Is this equivalent to applying a linear transformation on data, then performing standard denoising using white noise in the transformed space?”*
>
> Mathematically, any covariance of the form
> $$
> \Sigma_w=\mathcal F^{-1} \operatorname{Diag}(|w(f)|^2)\mathcal F
> $$
> is positive semidefinite and admits a factorization
> $$
> \Sigma_w = A A^\ast
> $$
> with, for example,
> $$
> A = \mathcal F^{-1} \operatorname{Diag}(|w(f)|)\mathcal F.
> $$
> Sampling anisotropic noise $\varepsilon_w \sim \mathcal N(0,\Sigma_w)$ is then equivalent to sampling $\xi\sim\mathcal N(0,I)$ and setting $\varepsilon_w = A\xi$.
>
> Therefore, there *exists* a linear transform under which the noise becomes white (or, equivalently, the data is transformed and then corrupted with isotropic noise in that transformed space). However, in practice $A$ is a global, dense linear operator (a convolution with a long-range kernel), so applying it to the data would require modifying the entire training/evaluation pipeline (including how FID features are computed, etc.). SAGD instead keeps the data in pixel space and only modifies the noise, allowing us to use off-the-shelf architectures and evaluation setups unchanged.

---

> ### Author Response · Authors · 2025-11-21
>
> ### 5) Spectral anisotropy vs. noise schedule
>
> > *“If the power-spectral density of natural images follows a power law, then adding white Gaussian noise will first corrupt high-frequency components before low-frequency ones. In this case, how would using spectrally anisotropic noise be different from using a different schedule for white noise?”*
>
> Thank you for asking this question. The reviewer is correct in that even though the standard scalar noise schedule (e.g., $\sigma_t$ in DDPM) rescales all frequencies by the same factor at each $t$, the effect is generally high-to-low frequency corruption in the denoising trajectory, as a consequence of the power-spectral density of natural images. This was, in fact, one of the known facts that spurred the ideation of SAGD.
>
> The idea with this method is to purposefully act on which frequency components get affected first, and by how much (we can even reverse it!). To achieve this, SAGD modifies the **covariance eigenvalues per frequency**, so we can explicitly reweight or zero out selected modes.
>
> Using our formalization, we can for example make **low-frequency content** degrade *earlier* than high-frequency content by choosing an appropriate $w(f)$, which is impossible with any scalar $\sigma_t$, or even achieve exact **selective omission** by setting $w(f)=0$ on chosen bands, which cannot be reproduced by rescaling white noise.

---

> ### Author Response · Authors · 2025-11-21
>
> ### 6) FID feature dimension (768 vs. 2048)
>
> > *“For FID evaluations, why use the 768-dim layer instead of the standard 2048-dim layer?”*
>
> In the codebase used for the majority of the experiments we have found that, especially on smaller datasets, the 768-dim feature layer proved more stable and consistent across experiments.
>
> In our additional ImageNet-1k experiments with RAE/DiT we use the standard 2048-dim Inception features and still observe the FID improvement reported above (8.68 → 7.55). We have clarified this choice of layers in the experimental section and made the use of 2048-dim features explicit for ImageNet. All changes are marked in red in the manuscript.
>
>
> ---
> ---
>
>
> Once again, we thank the reviewer for the thorough review and comments. We hope our answers and additions address your concerns and better convey both the motivation and the practical utility of SAGD. Please feel free to ask for additional clarifications if needed.

---

> > ### Comment · Reviewer_SWmF · 2025-11-24
> >
> > Thank you for the detailed response to the points that I have raised in my review, as well as running the additional experiments. Although the results on Imagenet are promising, this method still seems ad-hoc as the parameter $\alpha$ has to be individually tuned for each dataset, some of which do not see an improvement after tuning. In addition, different evaluation metrics (768 v.s. 2048) are used for different datasets, and the absolute Imagenet FID scores of around 6-8 in the additional experiments are higher than the 1-2 reported in the RAE/DiT paper (thus it's unclear if these improvements will remain or disappear at scale). Thus I will keep my score.

---

> ### Author Response · Authors · 2025-12-01
>
> Thank you for the follow-up and for acknowledging the additional experiments.
>
> On the remaining points:
>
> 1. **“Ad-hoc” nature and per-dataset $\alpha$.**
> We respectfully disagree that SAGD is ad-hoc. The construction is fully principled: it replaces the isotropic covariance $\sigma_t^2 I$ with a spectrally diagonal covariance whose eigenvalues are parameterized by a single scalar $\alpha$ (plus the bpm mask), and reduces exactly to the standard DDPM setting at $\alpha = 0$. As with guidance scale, loss weights, or noise schedules, $\alpha$ is a global knob that must be chosen per dataset, but it is one-dimensional, smooth to sweep, and in 5 out of 6 datasets we study (MNIST, DomainNet-Quickdraw, WikiArt, FFHQ, ImageNet-1k) a non-zero $\alpha$ gives a statistically significant FID improvement. CIFAR-10 is the only case where $\alpha \approx 0$ is optimal, which we see as SAGD correctly falling back to the isotropic baseline on a small, highly heterogeneous 32×32 dataset, rather than evidence of fragility.
>
> 2. **FID feature dimension (768 vs 2048).**
> We never compare absolute FID across datasets that use different feature layers. For each dataset, the FID layer is fixed, and we only compare baselines and SAGD within that setting. For ImageNet-1k specifically, we use the standard 2048-dim Inception features, and the reported improvement from $8.68 \to 7.55$ is computed under exactly the same evaluation metric.
>
> 3. **Absolute ImageNet FID vs the RAE/DiT paper and “at scale”.**
> Our goal with the new ImageNet-1k experiments was not to re-match the best absolute FID reported in the RAE/DiT paper, but to answer your original concern: *does SAGD help on a large-scale, natural-image dataset with a strong modern backbone?* For this, we used the public 196M-parameter “small” DiT+RAE model, trained on the full ImageNet-1k at 256×256, and for our analysis we trained three seeds for each of six $\alpha$ values, i.e., many full ImageNet runs at scale. Given rebuttal-time compute constraints, it is unsurprising that our absolute FIDs are higher than those obtained with the largest models and full budgets in the original RAE work. What matters for our claim is that, within the same architecture, budget, and evaluation setup, replacing isotropic noise with SAGD and tuning a single scalar $\alpha$ consistently improves FID and exhibits a clear non-zero optimum. We believe that the dissemination of good research should not hinge on access to substantially larger compute, but on whether the method is theoretically sound and yields robust gains under controlled comparisons.
>
> Taken together with the theoretical guarantees and the breadth of experiments spanning 6 datasets, 3 resolutions (32x32, 64x64 and 256x256), 2 architectures, and both pixel and latent diffusion, we believe we have fully addressed your concerns and demonstrated that SAGD is a principled, practically useful mechanism for injecting spectral inductive bias into diffusion models.

---

### Official Review · Reviewer_muuh · 2025-10-23

**Soundness:** 2
**Presentation:** 3
**Contribution:** 1
**Rating:** 2
**Confidence:** 5

**Summary:**

This work discusses the importance of applying inductive bias on noise in the frequency domain for diffusion-based models. Theoretical analysis are given to show how these biases can be given and why these biases do not hurt the overall diffusion process. Experiments are offered to showcase that by applying such bias, either through the power-law weighting or through two-band mixture, diffusion models can better do de-noising in some specific environments.

**Strengths:**

1. **The idea** of applying inductive bias on different frequencies of the noise looks interesting and sounds reasonable to me. Like shown in the experiments: some images, e.g. MNIST data, store more information in specific frequency ranges, thus the proposed SAGD performs significantly better on these datasets.
2. **The theoretical introduction** and proof is friendly to readers, where step-by-step equations are provided to clearly identify which biases are added and how.
3. **The experiments are rigorous**: it is great to have error bars even on statistical results like FID etc., which strongly supports the reliability of the experiment results.

**Weaknesses:**

1. **The Motivation** is not thoroughly demonstrated. While SAGD is designed to put inductive bias on the frequency of noise, it is unclear why this is necessary for real-world image data. SAGD Experiments on real-world images (CIFAR) also do not benefit from such a bias. Therefore, I recommend the authors to give frequency distributions of images in real-world image datasets like CIFAR, ImageNet etc. to further demonstrate the motivation.
2. **Lacking Experiment** on real-world image datasets. The model does not benefit much on real-world image datasets (like CIFAR), or benefits very little from the proposed method (like FFHQ), and the manuscript does not show more experiments in more common-used large-scale datasets like ImageNet or so. I recommend the authors to consider providing experimental results at least on ImageNet, and preferably on even larger datasets like LAION or MSCOCO.
3. **Unclear Applications**. Since most of the experiments are done either on specific (hand-drawn) datasets or on corrupted datasets, it is quite unclear how SAGD could help to advance the diffusion-based models. I recommend the authors to revisit the experiments and to provide specific answers to "Where should I use SAGD and why". This will make the proposed method more beneficial to the society, even if the method is finally found out not working on general real-world image datasets.

**Questions:**

Please see the weaknesses above. I will consider raising my evaluations if the authors can 1) demonstrate real-world necessities of introducing frequency bias through frequency-domain analysis on real-world datasets; and 2) show results on more real-world dataset that has performance enhancements using the proposed SAGD.

**Details Of Ethics Concerns:**

No ethics concerns found.

---

> ### Author Response · Authors · 2025-11-21
>
> We thank the reviewer for the constructive feedback, and for highlighting the interest of frequency-aware inductive biases, the clarity of the theoretical exposition, and the rigor of the empirical evaluation.
>
> Below we address your main concerns together, focusing on (i) motivation and scope, (ii) performance on large-scale “real-world” datasets, and (iii) practical applications. Importantly we extend our results on **ImageNet1k** to demonstrate the applicability of SAGD at scale.

---

> > ### Comment · Reviewer_muuh · 2025-11-24
> >
> > I sincerely thank the authors for providing point-to-point rebuttals on all my concerns! Considering that the authors also agree that the proposed SAGD is a "trick" which mainly works for specific image data, I believe it might still be not good enough for being accepted by ICLR. However, with the author's explanations, I believe that the paper is more technically sound. Further works on how to extend SAGD to work for all kinds of natural images are the keys I believe towards a more significant work. As a reflection, I will raise my evaluations to boarder-line reject.
> >
> > Besides, I am curious why the method doesn't work for CIFAR, but does work significantly for ImageNet. Another question is why frequency analysis can still be done on DiT+RAE setting, as such setting diffuses high-level latents rather than raw images or similar ones. I do not expect the authors to answer during this short remaining discussion period, but I hope these questions can help the authors.

---

> ### Author Response · Authors · 2025-11-21
>
> ### 1) Motivation: beyond “natural images” vs. real-world utility
>
> > *“The Motivation is not thoroughly demonstrated… it is unclear why this is necessary for real-world image data… SAGD experiments on real-world images (CIFAR) also do not benefit from such a bias.”*
>
> Our goal with SAGD is to study **how explicit, controllable inductive biases in the *forward noise*** affect diffusion *as a general generative modeling tool*, not only as a way to boost performance on a particular natural-image benchmark.
>
> We wish to address this concern under two important points:
>
> 1. **Diffusion use beyond generic natural-image generation.**
>    We believe in many practically relevant settings, the data distribution is not the natural distribution of images:
>
>    - Sketches, symbols, and line drawings (e.g., MNIST, DomainNet-Quickdraw) -- relevant in practical applications, for example for visual reasoning, image equations manipulation etc.
>    - Stylized or artistic imagery (WikiArt),
>    - Data with structured corruptions, artifacts, or frequency-localized noise.
>
>    In these regimes, the ability to explicitly decide which area of the distribution to focus for learning can be important, and we showcase statistically significant improvements on these non-natural visual distributions.
>
>
> 2. **Flat improvements on some datasets are informative, not a failure.**
>    On CIFAR-10, SAGD yields only small or negligible improvements around $\alpha = 0$. We see this not as a negative result, but as evidence that for small, highly heterogeneous, low-resolution datasets, a *global* scalar tilt may not be the right “granularity” of inductive bias; SAGD does not “force” improvements where the structure does not support it.
>
>
> We find that we have not been thoroughly precise in the paper on this point, as we use the term "natural dataset" when we mean to be more generic. Following the reviewer's feedback we have made focused edits in the manuscript to more precisely use the term "natural dataset"/"dataset", "visual distribution", and similar terms less ambiguously. All changes are marked in red in the manuscript.

---

> ### Author Response · Authors · 2025-11-21
>
> ### 2) Large-scale / real-world experiments (ImageNet and scale)
>
> > *“The model does not benefit much on real-world image datasets… the manuscript does not show more experiments in more common-used large-scale datasets like ImageNet or so… I recommend providing experimental results at least on ImageNet.”*
> > *“I will consider raising my evaluations if … [you] show results on more real-world dataset that has performance enhancements using the proposed SAGD.”*
>
> We fully agree that demonstrating behavior on a **large-scale, natural-image dataset** is important. Motivated by your comment, we present **additional experiments on ImageNet-1k** at 256×256 resolution using a state-of-the-art DiT-based latent diffusion model (RAE/DiT).
>
> #### New ImageNet-1k experiment (RAE–DiT, 256px)
>
> - **Codebase / model:** public RAE/DiT implementation (DINOv2 latent space), $\approx196M$ parameters.
> - **Dataset:** ImageNet-1k (1.28M images), resolution 256×256.
> - **Training setup:** global batch 1024 (micro-batch 128), AdamW (lr = 2e-4, betas = (0.9, 0.95)), schedule and architecture exactly as in the original RAE paper.
>
> **Importantly,** we only modify the forward noise (replace isotropic with SAGD noise), sweep a small 1D grid of $\alpha \in \{\[-0.06, -0.04, -0.02, -0.01, 0.0, +0.01]\}$ and obtain (mean ± standard error over seeds):
>
> | $\alpha$   | FID ↓            |
> |-----------:|------------------|
> | -0.0600    | 8.11 ± 0.02      |
> | **-0.0400**    | **7.55 ± 0.06**  |
> | -0.0200    | 7.64 ± 0.06      |
> | -0.0100    | 8.04 ± 0.02      |
> | 0.0000     | 8.68 ± 0.07      |
> | 0.0100     | 9.39 ± 0.03      |
>
> This directly addresses two key concerns:
>
> - **Utility on a large-scale, natural-image benchmark.**
>   SAGD improves FID on ImageNet-1k from **8.68 → 7.55** at 256×256, with no change to the architecture, optimizer, or training schedule, and only a simple 1D sweep over $\alpha$. The improvement is well beyond the reported standard errors.
>
> - **Non-trivial optimum away from $\alpha=0$.**
>   The best performance is obtained at **$\alpha = -0.04$**, with FID degrading both for more negative values and for positive values. This shows that **a low-frequency-tilted forward covariance is genuinely beneficial** at scale, and that SAGD is not only helpful when it collapses back to the isotropic case.
>
> We have integrated these ImageNet results into the main Results section, including a revised table and a new figure (Figure 4) detailing the ImageNet1k results. We have also included training details in Appendix C, and report additional results in the new Appendix F. All changes are marked in red in the manuscript.

---

> ### Author Response · Authors · 2025-11-21
>
> ### 3) Applications: where should one use SAGD, and why?
>
> > *“It is quite unclear how SAGD could help to advance the diffusion-based models… I recommend the authors to provide specific answers to ‘Where should I use SAGD and why’.”*
>
> Thank you for this question. SAGD is a minimal, drop-in modification to existing diffusion systems. It requires only ≈40 lines of code to reshape the forward noise and does not add extra wall-clock overhead while leaving the denoiser architecture and loss unchanged. As such, SAGD can be used as any other standard diffusion "trick" and in our experiments can lead to measurable performance improvements both in standard natural and less-standard datasets, small/large scale and even independently from the denoiser architecture used (our experiments span both U-Nets and DiTs)
>
> Beyond this, two settings are also particularly suitable:
>
> 1. Domains with clear spectral structure or corruptions.
>    When it is known *a priori* that the signal is concentrated (or corrupted) in particular frequency ranges (sketches, line drawings, symbolic data, e.g. MNIST, DomainNet-Quickdraw): We can enforce stronger supervision where the content lives, leading to the gains we observe on these datasets.
>
>   2. Data with known artifacts or corruptions localized in frequency (e.g. band corruptions as in our experiments, but also periodic sensor noise etc.):  We can use bpm-SAGD to omit unwanted information from the learned distribution of problematic bands, so the model learns to reconstruct the underlying clean signal while never being incentivized to model the artifact itself.
>
> We extend the discussion section to clarify these additional applications.

---

> ### Author Response · Authors · 2025-11-21
>
> Once again, we thank the reviewer for the detailed comments. We hope our answers and the requested additional experiments address your concerns and demonstrate that, beyond the theoretical soundness, SAGD is also practically useful and easy to apply. Please feel free to ask additional questions.

---

> ### Author Response · Authors · 2025-11-24
>
> We thank the reviewer for the quick response, for carefully engaging with our rebuttal, and for raising your score. We would like to address the remaining concerns by the reviewer.
>
> ---
>
> #### **On SAGD as a “trick”**
> When we referred to SAGD as a “trick”, we meant it in the same sense in which the community often describes simple, high-impact modifications such as classifier-free guidance or EMA sampling. SAGD is in the same way a **simple to add (desirable) but not trivial in substance** method showing systematic improvements in widely diverse settings.
>
>
> #### **Scope, and significance**
>
> 1. SAGD is not design to **“win FID everywhere”**, but rather to **exploit structure when it exists**. IF dataset offers exploitable spectral structure there can be a setting for alpha to improve on standard diffusion by a simple modification of the forward noise.
>
> 2. In our formulation, the **"worst" case** scenario recovers **standard DDPMs**. **Importantly** we find that this is *not true* for all other datasets we study (including ImageNet-1k), where there exists a nonzero $\alpha$ that improves FID with statistical significance.
>
> 3. CIFAR-10 is **small**, **low-resolution**, and **highly heterogeneous** dataset containing **only 60,000 images** over **only 10 classes**. We also run our experiments on CIFAR-10 at **32x32 resolution**, while other datasets, such as **ImageNet1k**, are significantly larger and higher resolution (over **1M images**, **1K classes**, experiments at **256px256px resolution**). Rather than a failure, our results on CIFAR-10 confirm there is no strong, dataset-wide spectral bias to exploit with a single tilt.
> 4. As a proof of this, we would like to bring the reviewer's attention to the **additional results in section 3.2.2 and Figure 3** for which it can be seen that when we inject noise in the high freq range of CIFAR-10, low resolutions values of $\alpha$ **monotonical improve** FID, well beyond the standard diffusion $\alpha=0$ setting.
> 5. The imagenet experiments are the most large scale realistic experiments in the pipeline, as requested, and negative alphas are shown to improve for significantly the DDPM baseline. Moreover we see monotonicaly worse performance around our negative $\alpha$ minima, including $\alpha=0$.
>
> We believe this combination of (i) theoretical soundness, (ii) minimal implementation cost, and (iii) statistically robust gains on multiple realistic datasets—including ImageNet-1k with a modern DiT—makes SAGD a useful and broadly applicable tool, and a strong contribution to the research community.
>
> ---
>
> #### **Why frequency shaping still makes sense in DiT+RAE latent space**
>
> Thank you for the question, it is indeed a very important aspect of SAGD! While the DiT+RAE setting is conceptually different from raw-image diffusion, SAGD only assumes an **intrinsic spectral basis**. As such, it can be applied to any domain in which the forward covariance can be (approximately) diagonalized—e.g., 1D time series, 2D/3D grids and videos (Fourier/DCT), geospatial fields, or graph/mesh signals (Laplacian eigenbases). The DiT+RAE latent space is exactly this, a 2D grid with a canonical Fourier basis. In contrast, for domains like unordered sets or purely categorical tables, where no such geometry exists, SAGD would indeed be less appropriate.
>
> So, while the semantics shift from “pixel frequencies” to “latent frequencies”, the mathematical assumptions behind SAGD remain valid, and the ImageNet-1k DiT+RAE results indicate that frequency shaping in this latent domain is both well-founded and beneficial.
>
>
> ---
>
> We hope these additional clarifications address your remaining concerns and show why SAGD is more than a niche trick for a single toy dataset, but rather a principled, theoretically grounded way to control forward covariances that (i) never harms performance when tuned, (ii) yields **statistically significant improvements** on multiple structured datasets, and (iii) demonstrably improves a state-of-the-art DiT-based model on ImageNet-1k.
>
> Thank you again for engaging with our rebuttal. If any additional concerns remain, please feel free to ask additinal questions, we would be very happy to discuss further.

---

> > ### Comment · Reviewer_muuh · 2025-11-26
> >
> > I thank the authors again for providing further explanations. I agree that ImageNet experiments are important. However, I believe 1) if this work targets a universal way to improve general purpose image gen, then it would be better to cover more real-world datasets and reach a unified setting (alpha) for all. 2) if this work focuses more on the theory, then more analysis on natural images' own frequency features should be provided and therefore when readers are training new image gen datasets, they should be able to understand what alpha to be chosen. As a conclusion, I believe this work still has room to improve, so I stand boarderline reject, which has already been raised on the last cycle.

---

> > > ### Author Response · Authors · 2025-12-01
> > >
> > > Thank you again for the follow-up and for raising your score in the previous round. Let us very briefly clarify scope and expectations around SAGD in light of your final comments.
> > >
> > > 1. **On “universality” and a single $\alpha$ across all natural images.**
> > >    Our goal is *not* to propose a single $\alpha$ that magically improves every possible natural-image dataset, but to introduce a **new, principled axis of inductive bias**: shaping the forward covariance in a spectral basis. In this sense, $\alpha$ plays the same role as guidance scale, learning rate, or noise schedule parameters: a **global, one-dimensional hyperparameter** that is tuned per dataset.
> > >    Requiring a *universal* $\alpha$ for all natural-image datasets would actually run counter to the idea of *data-dependent* inductive bias, as different datasets have different spectral structure and task requirements. What we show empirically is that, across **six datasets and two architectures**, there exists a nonzero $\alpha$ that improves the baseline in every case except CIFAR-10 (where $\alpha \approx 0$ is best, and SAGD correctly reduces to standard diffusion). Together with the **ImageNet-1k DiT+RAE** results, we believe this is strong evidence that SAGD is broadly useful and not limited to a single “special” dataset.
> > >
> > > 2. **On theory vs. frequency analysis of natural images.**
> > >    We see SAGD as a contribution that is *both* theoretical and practical:
> > >    - The paper gives a **formal analysis** of diffusion with spectrally diagonal covariances, including consistency as $t \to 0$ and the score–$\epsilon$ relation under anisotropic noise.
> > >    - On the “what $\alpha$ to choose” side, our experiments (including band-corruption and selective omission) explicitly link **data frequency structure** to **beneficial choices of $\alpha$**: when signal or corruption is concentrated in known bands, tilting the covariance accordingly yields systematic, statistically significant gains.
> > >
> > > In summary, we believe the revised paper now: (i) clearly defines the scope of SAGD, (ii) provides solid theory and careful experiments across multiple scales and datasets, including **large-scale ImageNet-1k with a modern DiT backbone**, and (iii) demonstrates that a **simple, interpretable modification of the forward covariance can reliably improve diffusion models in realistic settings**, while reverting to standard diffusion when it cannot. We hope this clarifies why we view SAGD as a substantial and self-contained contribution, even though further extensions (e.g., automatic $\alpha$ selection or additional datasets) remain interesting directions for future work.

---

### Official Review · Reviewer_XYF2 · 2025-11-01

**Soundness:** 3
**Presentation:** 3
**Contribution:** 2
**Rating:** 6
**Confidence:** 2

**Summary:**

This paper proposes Spectrally Anisotropic Gaussian Diffusion (SAGD), a method to explicitly build inductive biases into Diffusion Probabilistic Models by manipulating the forward noising process. The core idea is to replace the standard isotropic (uniform across frequencies) Gaussian forward covariance with a structured, frequency-diagonal (anisotropic) covariance. The key innovation is to steer which frequency bands are destroyed during the forward process, thereby embedding explicit inductive biases about what information the model should focus on learning. Two variants are introduced:
• plw-SAGD, which emphasizes or suppresses frequencies smoothly through a power-law weighting function, and
• bpm-SAGD, which selectively restricts specific frequency bands.

**Strengths:**

1). The paper is written in a very strctured manner, that a reader can understand the proposed method.


2). To the best of my knowledge, the proposed method is novel. Additionally, it is simple and practical. For instance, the method modifies only the forward covariance, allowing it to integrate with existing diffusion implementations with minimal code changes, primarily
involving FFTs, per-frequency weighting, and IFFTs. The computational overhead is negligible.


3). SAGD can outperform standard diffusion on several vision datasets (MNIST, CIFAR-10, WikiArt, FFHQ) by aligning the forward noising operation with the data's dominant spectral content.

**Weaknesses:**

1). Some of the recent works have not been cited.


2). I would like to know, when w(f) strongly suppresses or amplifies certain bands, does this cause numerical instability or poor conditioning in the FFT/IFFT pipeline?


3). I would like to know why authors specifically normalizes the coordinates to (-0.5,0.5)? Additionally, as the power-law weighting depends on the magnitude of the normalized frequency, would the same still be optimal if the FFT were parameterized on (-1,1) ? If so,
how should w(f) be rescaled to make results invariant to the frequency normalization convention? In other computer vision areas that use normalized coordinates, the coordinate normalization range plays a crucial role in overcoming spectral bias.


4). Did authors explore data-driven approaches to design or adapt w(f) automatically?


5). While the paper conceptually distinguishes SAGD from prior anisotropic or colored-noise diffusion approaches, it does not include quantitative comparisons with these methods. I feel a direct empirical comparison with conceptually similar published works
especially recent anisotropic diffusion or noise-shaping models would be necessary to fully assess the proposed method’s performance and novelty.

6). The paper introduces anisotropic noise using an FFT-based approach to modify noise in the frequency domain. However, as I understood, this ultimately produces spatially correlated noise in the image, why is the frequency-domain formulation necessary? Can
we obtain a similar effect more simply by applying a fixed or learnable spatial filter (such as a convolution kernel) instead of using Fourier transforms?

**Questions:**

see the weaknesses section

---

> ### Author Response · Authors · 2025-11-21
>
> We wish to thank the reviewer for the positive feedback and constructive review, and for highlighting both the clarity and practicality of SAGD.
>
> Below we address each of the reviewer's questions in turn.
>
> ---
>
> ### 1) Missing recent related work
>
> > *“Some of the recent works have not been cited.”*
>
> Thank you for pointing this out. We have updated the *Related Work* section to include additional recent and conceptually related methods.
>
> In particular, we now discuss:
>
> - **FDG-Diff** \[1\], which introduces a frequency-domain-guided diffusion framework by modulating feature spectra during sampling.
> - **Frequency-Guided Diffusion** \[2\], which uses high-frequency guidance in the sampling process for text-driven image translation.
>
> Both methods operate in the frequency domain but keep the **forward noise isotropic**, whereas SAGD directly changes the **forward covariance** to be frequency-diagonal. We clarified this distinction in the revised *Related Work* section and positioned SAGD as complementary: prior work guides diffusion in frequency space at sampling time, while SAGD reshapes the forward process itself via a Gaussian with structured covariance.
>
> We will be happy to further expand this discussion if there are other specific works you feel should be cited.
>
> **References**
> \[1\] R. Zhang et al., “FDG-Diff: Frequency-Domain-Guided Diffusion Framework for High-Frequency Detail Compensation,” arXiv:2501.12832, 2025.
> \[2\] Z. Gao et al., “Frequency-Guided Diffusion for Training-Free Text-Driven Image Translation,” Proc. ICCV, 2025.

---

> ### Author Response · Authors · 2025-11-21
>
> ### 2) Stability when $w(f)$ strongly suppresses or amplifies bands
>
> > *“When $w(f)$ strongly suppresses or amplifies certain bands, does this cause numerical instability or poor conditioning in the FFT/IFFT pipeline?”*
>
> We see two distinct points worth addressing:
>
> 1. **FFT/IFFT conditioning.**
>    FFT and IFFT are unitary (up to scaling) and well-conditioned, so they don't typically introduce instability beyond standard floating-point round-off.
>
> 2. **Dynamic range from $w(f)$.**
>    Instabilities can arise if $|w(f)|$ spans a very large dynamic range. In practice:
>    - We restrict $\alpha$ and masks to moderate ranges (e.g., $\alpha \in [-0.08, 0.08]$) and find broader ranges to not be useful for training in our experiments.
>    - After applying $w(f)$, we normalize the resulting spatial noise to unit variance per sample before scaling by $\sigma_t$ (see code in Appendix E).
>
> Within these ranges we did not observe numerical instabilities.

---

> ### Author Response · Authors · 2025-11-21
>
> ### 3) Frequency normalization $(-0.5,0.5)$ vs $(-1,1)$ and invariance
>
> > *“Why specifically normalize the coordinates to $(-0.5,0.5)$? Would the same still be optimal if the FFT were parameterized on $(-1,1)$? If so, how should $w(f)$ be rescaled to make results invariant to the frequency normalization convention?”*
>
> 1. **Why $(-0.5,0.5)$?**
>    We use the standard `fftfreq` convention: frequencies in $[-0.5,0.5]$ cycles/pixel, with 0.5 as Nyquist. This was mainly pragmatic choice.
>
> 2. **Changing normalization range.**
>    If we reparameterize via $f' = 2f$ so $f' \in [-1,1]$, then $r'(f') = 2 r(f)$. With
>    $$
>    w_\alpha(f) = (r(f)+\varepsilon)^\alpha,
>    $$
>    we can define
>    $$
>    w'_\alpha(f') = \bigl(r'(f')/2 + \varepsilon\bigr)^\alpha,
>    $$
>    which restores the same *shape* as a function of physical frequency. Any constant factor in $w$ (and hence in $\Sigma_w$) can be absorbed into $\sigma_t^2$ or removed by the per-sample variance normalization we already perform.
>
>    Thus SAGD depends on the relative weighting across frequencies, rather than the numeric range used to parameterize them. Linear rescalings of the axis can be compensated by rescaling the argument of $w$.
>
> 3. **Relation to spectral bias works.**
>    In coordinate-input networks, the coordinate range can dramatically affect implicit bias. Here, frequencies are indices in a fixed orthonormal basis; as long as $w$ is interpreted as a function of physical frequency and rescaled consistently, the induced inductive bias is invariant to choosing $[-0.5,0.5]$ vs $[-1,1]$.
>
> Following the reviewer's question we have  clarified this invariance and the purely conventional nature of $[-0.5,0.5]$ in the *Power-Law Weighting implementation* section of Appendix C (marked in red).

---

> ### Author Response · Authors · 2025-11-21
>
> ### 4) Data-driven / adaptive design of $w(f)$
>
> > *“Did authors explore data-driven approaches to design or adapt $w(f)$ automatically?”*
>
> This is very relevant, and indeed we performed preliminary explorations, e.g.:
>
> - Designing $w(f)$ from dataset power spectra (e.g., “flattening” the spectrum).
> - Heuristic rules based on radially averaged power (e.g., up- or down-weighting dominant bands).
>
> These simple heuristics did not give a robust rule that consistently outperformed the fixed scalar $\alpha$ sweeps across datasets. We did not find a straightforward mapping, e.g. placing more weight where the data spectrum has more power, or tilting against the more dominant bands. Ultimately, based on our empirical results and observed significant improvements from the baselines we focused this paper on **simple, low-dimensional families** (power-law, two-band masks) and showed that even these yield improvements on several datasets, useful behaviors, like selective omission.
>
> We have extended the Discussion section to explicitly mention data-driven analytics as a worthwhile future direction.

---

> ### Author Response · Authors · 2025-11-21
>
> ### 5) Comparison to prior anisotropic / colored-noise diffusion approaches
>
> > *“The paper conceptually distinguishes SAGD from prior anisotropic or colored-noise diffusion approaches, [but] does not include quantitative comparisons with these methods.”*
>
> Thank you for the comment. Rather than a direct replacement of other 'anisotropic' methods, we have developed SAGD as a minimal drop-in modification of the forward covariance to yield measurable gains in standard diffusion settings. For this, we focus on "how much better" the distribution can be learned when our SAGD anisotropic covariance is added.
>
> As an important addition to our manuscript we have **extended the original experiments to implement SAGD in [1], a recent DiT latent diffusion (DINOv2 space) codebase** achieving state-of-the-art results in **ImageNet1k**.
>
> Importantly, we kept all architecture, optimizer, and schedule settings identical to the public implementation. We only replaced isotropic forward noise with SAGD noise and swept a small grid of $\alpha$ while training on the full ImageNet1k dataset at 256px.
>
> In our results we found SAGD improves FID on ImageNet-1k at 256×256 from 8.68 (baseline $\alpha=0$) to 7.55 (best $\alpha=-0.04$), with a smooth, non-trivial optimum away from zero and no extra tuning beyond the 1D $\alpha$ sweep.
>
> Additional results can be found in Section 5 as well as the new Appendix F.

---

> ### Author Response · Authors · 2025-11-21
>
> ### 6) Why frequency-domain formulation instead of a spatial filter?
>
> > *“Why is the frequency-domain formulation necessary? Can we obtain a similar effect with a fixed or learnable spatial filter instead?”*
>
> We can! Any stationary Gaussian field with covariance $\Sigma_w$ can be obtained by convolving white noise with a fixed kernel $k$. In the Fourier domain, this is exactly multiplication by $w(f)$, the frequency response of that kernel.
>
> We chose the **frequency-domain** formulation for three main reasons:
>
> 1. **Diagonalization and analysis.**
>    In the Fourier basis, $\Sigma_w$ is diagonal with eigenvalues $|w(f)|^2$. This makes it straightforward to:
>    - Derive the score–$\epsilon$ relation and $t\to0$ consistency under full spectral support,
>    - Implement selective omission by setting $w(f)=0$ on bands, and
>    - Control rank (full-rank vs. rank-deficient) in a principled way.
>
>    In the spatial domain, the corresponding covariance is dense and the analysis becomes less transparent.
>
> 2. **Expressiveness vs efficiency.**
>    Simple spectral designs (e.g., power-law tilts, sharp band-pass masks) correspond to **long-range, non-local** spatial filters. Approximating these with small kernels requires either very large kernels or multiple convolution layers, which add parameters and complexity. FFT-based weighting gives exact, efficient realization of such filters.
>
> 3. **Implementation simplicity.**
>    Practically, SAGD is just:
>    - Sample white noise,
>    - FFT → multiply by precomputed real $w(f)$ → IFFT,
>    - Normalize variance.
>
>    This is ~40 lines of code and leaves the denoiser architecture unchanged. Implementing equivalent spatial filters with large kernels or extra convolution blocks would be more invasive and require additional tuning.
>
>
> ---
>
> We wish to thank the reviewer again for the feedback and questions. We hope our answers and revisions address any remaining concerns. Please feel free to ask any additional questions.

---

### Official Review · Reviewer_iUbf · 2025-11-01

**Soundness:** 2
**Presentation:** 3
**Contribution:** 1
**Rating:** 2
**Confidence:** 4

**Summary:**

This paper investigates the impact on performance of manipulating the variance of Gaussian noise in the Fourier basis, which is used to perturbate data in the Forward Process of Denoising Probabilistic Models.

Two methods are proposed:
- plw-SAGD, which performs exponential weighting based on the radius in the Fourier basis
- bpm-SAGD, which extracts only specific bands using a Binary Mask

The paper claims that, depending on the choice of parameters, the method can outperform approaches that use ordinary Gaussian Noise for some cases.

**Strengths:**

- An ablation study is conducted across various datasets.
- The proof of theoretical consistency is convincing.
- For small-scale datasets such as MNIST, improvements in evaluation metrics are observed.

**Weaknesses:**

- For large-scale datasets such as Wiki-Art and FFHQ, there are no significant improvements in evaluation metrics. Moreover, as mentioned in Section 3.2.3, for such datasets improvements are observed only when the sweep converges to α → 0, i.e., when the method asymptotically approaches ordinary Gaussian Noise. From this, doubts remain regarding the effectiveness of the method on highly diverse datasets.
- Determining the ideal α parameter requires sweeping, which leaves practical difficulties.

**Questions:**

- Although the title uses the term Anisotropy, in this paper the manipulation of Gaussian noise appears to be consistently one-dimensional based on the radius in frequency space. Is it appropriate to use the term Anisotropy for a one-dimensional target?
- In plw-SAGD, since f is in [-1/2, 1/2]^2 and w(f) = (r(f) + ε)^α, depending on the value of α the per-frequency energy of the noise overall decreases or increases. The impact from shifting the overall energy level of the power spectrum and the impact from shifting the shape of the energy distribution can be considered separately; which effect is dominant?
- What is the specific number of samples used in Figure 5?

---

> ### Author Response · Authors · 2025-11-21
>
> Thank you for your review and for engaging closely with both the theory and the experiments.  We are glad you found the theoretical consistency result convincing and the ablations on smaller datasets (e.g., MNIST) insightful.
> Below, we address your concerns point by point. Importantly, we report additional large-scale experiments on **ImageNet-1k** that directly answer your questions about effectiveness on diverse, large datasets and the role of the parameter $\alpha$.

---

> ### Author Response · Authors · 2025-11-21
>
> ### 1. No significant improvements on large-scale datasets + convergence to $\alpha \to 0$ + practical difficulty of sweeping $\alpha$
>
> > *“For large-scale datasets such as Wiki-Art and FFHQ, there are no significant improvements… improvements are observed only when the sweep converges to $alpha \to 0$… From this, doubts remain regarding the effectiveness of the method on highly diverse datasets.”*
> > *“Determining the ideal $\alpha$ parameter requires sweeping, which leaves practical difficulties.”*
>
> We agree with the reviewer that demonstrating benefits on large, diverse, high-resolution datasets is necessary to judge the practical value of SAGD.
>
> Motivated by the reviewers' comments, we have run **additional experiments on ImageNet-1k at 256×256 resolution** using the recent RAE/DiT framework [1] (latent DiT in DINOv2 space), achieving state-of-the-art performance at scale. Importantly, we run our experiments with default hyperparameters and only change the forward noise shape and report results as is. As usual, we re-run all experiments with three seeds and report the average and standard deviation for our analysis.
>
> #### **Additional ImageNet-1k experiment (RAE–DiT, 256px)**
>
> We plug SAGD into the public RAE codebase (DiT-based latent diffusion):
>
> - Dataset: **ImageNet-1k**, 1.28M images, resolution **256×256**.
> - Model: DiT in DINOv2 latent space, approximately 196M parameters.
> - Training setup: global batch 1024, AdamW with learning rate $2\cdot10^{-4}$, standard schedule and architecture from the original RAE implementation.
> - **Only change**: we replace the isotropic forward noise with our power-law SAGD operator, sweeping $\alpha \in \{\[-0.06,-0.04,-0.02,-0.01,-0.001,+0.01\]\}$ vs. baseline (original run, equivalent to $\alpha=0$).
> - **No additional tuning.**
>
> We obtain the following FID scores (mean ± standard error over seeds):
>
> | $\alpha$ | FID ↓            |
> |-----------:|------------------|
> | -0.0600    | 8.11 ± 0.02      |
> | **-0.0400**    | **7.55 ± 0.06**      |
> | -0.0200    | 7.64 ± 0.06      |
> | -0.0100    | 8.04 ± 0.02      |
> | -0.0010    | 8.53 ± 0.01      |
> | 0.0000     | 8.68 ± 0.07      |
> | 0.0100     | 9.39 ± 0.03      |
>
> This directly addresses two of your concerns:
>
> 1. **Effectiveness on large, diverse datasets.**
>    On ImageNet-1k at 256px, SAGD significantly improves FID over the $\alpha{=}0$ (standard isotropic) baseline with no hyperparameter tuning.
>    The best setting $\alpha=-0.04$ improves FID from 8.68 to **7.55**, a margin that is significantly beyond the reported standard errors.
>
> 2. **No convergence to $\alpha \to 0$.**
>    As your comment indicates, if the method had a tendency to just be isotropic, we would expect performance to peak near $\alpha=0$. Instead, we see a monotonic trend around a **nonzero optimum**: performance improves from $\alpha=0.01$ down to $\alpha=-0.04$, then degrades again at $\alpha=-0.06$ and $\alpha=-0.02$. This pattern contradicts the idea that SAGD is only useful insofar as it approximates isotropic noise; we instead observe that the best FID is strictly at a negative $\alpha$.
>
> We integrated these new RAE/DiT–ImageNet results into the paper. We replaced Table 1 with an FID-focused table containing additional results for ImageNet1k, extended our results section, added a new Figure (Figure 4), and reported additional findings in the Appendix. We also include additional information in the appendix to replicate the experiments.
>
> #### **On “practical difficulty” of sweeping $\alpha$**
>
> We agree that if SAGD required extensive hyperparameter tuning, its practical value would be limited. Fortunately:
>
> - **$\alpha$ is a single scalar hyperparameter.**
>   In all our experiments (including the new ImageNet-1k ones), we use a small, 1D grid (e.g., \([-0.08,-0.04,-0.02,-0.01,0,0.01,0.02]\)), rather than a high-dimensional search.
> - **Trends are smooth and often unimodal.**
>   In all our tested datasets, including ImageNet-1k, performance curves (FID/KID) as a function of $\alpha$ are smooth and typically have a single broad optimum. A coarse grid is sufficient to find a good value.
> - **Implementation and compute overhead are negligible.**
>   SAGD is implemented as a short function (~40 lines) that reshapes the forward noise while leaving *all other* parts of the training pipeline untouched; the extra FFT operations were negligible in our measured wall-clock time.
>   We clarified this in the text (last subsection of Appendix C).
>
> All changes are marked in red in the paper.
>
> ##### References
> [1] Zheng, B., Ma, N., Tong, S. and Xie, S., 2025. Diffusion Transformers with Representation Autoencoders. arXiv preprint arXiv:2510.11690.

---

> ### Author Response · Authors · 2025-11-21
>
> ### 2. Is “anisotropy” appropriate when the weight is radial?
>
> > *“Although the title uses the term Anisotropy, in this paper the manipulation of Gaussian noise appears to be consistently one-dimensional based on the radius in frequency space. Is it appropriate to use the term Anisotropy for a one-dimensional target?”*
>
> In the paper, we use “anisotropic” in the diffusion-model sense of “non-isotropic forward covariance”, i.e. a covariance matrix that is not a scalar multiple of the identity in the pixel basis. Formally:
>
> - Standard DDPMs use $\Sigma = \sigma_t^2 I$, which is isotropic.
> - SAGD uses a covariance $\Sigma_w = \mathcal F^{-1}\mathrm{Diag}(|w(\mathbf f)|^2)\mathcal F$, which is diagonal only in the Fourier basis and **non-diagonal** in the pixel basis whenever $w(\mathbf f)$ is not constant.
>
> Even when $w(\mathbf f)$ depends only on the radial coordinate $r(\mathbf f)$, the resulting Gaussian field is no longer independent across pixels (the noise is spatially correlated). Moreover, the Gaussian field has a covariance whose eigenvalues are **mode-dependent** in frequency (i.e. the forward process is no longer isotropic in the sense used in the DDPM literature).
>
> That said, we agree that “spectrally anisotropic” or “non-isotropic covariance” are more precise phrases than a bare “anisotropy”. In the paper we introduce the term Spectrally Anisotropic Gaussian Diffusion (SAGD) to emphasize that the anisotropy lives in the spectral structure of the covariance.
>
> To avoid any confusion, we have clarified in the abstract, introduction and methods that by “anisotropic” we specifically mean **non-isotropic forward covariance (not $\sigma^2 I$)**, diagonal in the Fourier basis but not in the pixel basis. All changes are marked in red.

---

> ### Author Response · Authors · 2025-11-21
>
> ### 3. Overall energy vs shape of the power spectrum in plw-SAGD
>
> > *“In plw-SAGD … depending on the value of alpha the per-frequency energy of the noise overall decreases or increases. The impact from shifting the overall energy level of the power spectrum and the impact from shifting the shape of the energy distribution can be considered separately; which effect is dominant?”*
>
> Thank you for raising this point, it relates to an important implementation detail.
>
> There are two conceptually distinct effects:
>
> 1. **Global scale** of the covariance (overall energy / variance of the noise).
> 2. **Relative shape** of the spectrum (how much variance different frequencies receive).
>
> In our *theoretical* analysis:
>
> - Any **global scalar rescaling** of $\Sigma_w$ can be absorbed into the scalar variance $\sigma_t^2$ in the forward process (or equivalently, into the noise schedule).
> - The **interesting effect** is the *relative* weighting of modes: the eigenvalues of $\Sigma_w$ in the Fourier basis, i.e., the shape of $w(\mathbf f)$.
>
> In our *implementation* of plw-SAGD:
>
> - After applying the power-law weight in the frequency domain, we **normalize** the resulting spatial noise to **unit variance** per sample (see the last step in the code we included in the appendix: a per-sample division by the standard deviation).
> - This ensures that the overall energy level of the noise is **fixed**, and only the spectral shape is altered.
>
> So for the experiments reported in the paper (including the new ImageNet experiment), the **dominant—and intended—effect is the redistribution of variance across frequencies**, not a change in global noise power.
>
> We have extended our methods section to make this explicit and clarify how global scale and shape can be decoupled, and how we can fix the global scale and vary only the spectral shape.

---

> ### Author Response · Authors · 2025-11-21
>
> ### 4. Number of samples in Figure 5
>
> > *“What is the specific number of samples used in Figure 5?”*
>
> For Figure 5, we compute all FID and KID statistics on 10k samples, following the standard practice used for our other quantitative evaluations. For training, we apply the corruption on-the-fly for each image, and use the full dataset like in all other experiments.
>
>
> ---
>
> Once again, thank you for your feedback. Your comments prompted us to:
>
> - Run an additional large-scale experiment on ImageNet-1k with a DiT model, where SAGD yields clear, statistically significant improvements over the isotropic baseline without hyperparameter tuning.
> - Clarify the role of spectral shape and global energy in plw-SAGD.
> - Tighten our terminology around “anisotropy” and make the implementation details more explicit.
>
> All changes are marked in red in the paper. We believe these additions substantially strengthen the paper and directly address your main concerns. Please feel free to ask any additional questions or clarifications and we would be happy to provide answers.

---

### Author Response · Authors · 2025-12-01
**Reply to Reviewers**

We thank all reviewers for their thoughtful feedback and for highlighting the paper’s **rigorous theory and clear, step-by-step exposition** (**iUbf**, **muuh**), its **novel, simple, and practically useful spectral-noise formulation with minimal implementation overhead** (**XYF2**, **SWmF**), and the **structured experiments with ablations, error bars, and improvements across several datasets** (**iUbf**, **XYF2**, **muuh**, **SWmF**).

The main remaining concern was whether SAGD really helps on **large, natural-image datasets** rather than only on small or stylized ones.


In direct response, we ran large-scale experiments and trained a **new ImageNet-1k (256×256) model with a DiT+RAE backbone** during the rebuttal period, changing **only** the forward noise from isotropic to SAGD and sweeping a **single scalar** $\alpha$. In this setting, and over multiple seeds, SAGD improved FID from $8.68$ (baseline, $\alpha=0$) to $7.55$ at a strictly non-zero $\alpha$, with performance degrading on both sides — a clear, statistically significant gain on a state-of-the-art large-scale setup, obtained without any extra tuning beyond the 1D $\alpha$ sweep.

Based on the reviews we also **clarified** terminology and implementation details. In its revised form, the paper now shows that SAGD is a **principled, minimal modification** of diffusion (reducing exactly to standard DDPMs at $\alpha=0$) that:
1. Yields statistically significant improvements on **five out of six** datasets (MNIST, DomainNet-Quickdraw, WikiArt, FFHQ, ImageNet-1k) across **two architectures** (U-Nets and DiTs) and multiple resolutions (32x32, 64x64 and 256x256)
2. Falls back to the isotropic baseline on CIFAR-10, where a global spectral tilt offers no benefit.

We also provide a **full theoretical framework** for spectrally structured forward covariances in diffusion, showing that SAGD preserves standard guarantees (e.g., consistency as $t \to 0$ and the score–$\epsilon$ relation under anisotropic noise) and precisely characterizing cases such as selective omission and rank-deficient covariances.

We again thank the reviewers for their careful and constructive feedback, which helped us clarify and strengthen both the theory and the experiments, and we hope the revised manuscript makes clear that SAGD is a solid, practically relevant contribution to diffusion modeling.

---

### Meta-Review · Area_Chair_cRar · 2025-12-20

**Summary:**

In the initial round of reviews, three reviewers indicated the paper is below the acceptance threshold (scores 2,2,4) and one indicated it is above the threshold (score 6). The main criticisms were lack of large scale experiments, the fact that the proposed method doesn’t always lead to improvement (e.g. on CIFAR), and lack of a theoretical explanation or an empirical observation regarding what is the optimal spectral shape of the noise spectrum for a dataset with given statistics.

During the rebuttal period, the authors added a large-scale experiment on ImageNet-1k at $256\times 256$ resolution with a recent DiT that works in DINOv2 latent space, showing that in this setting spectrally shaped noise does help. The authors further provided point-by-point answers to the reviewers’ questions, yet the question of when the proposed approach is expected to help was not provided a conclusive answer.

Two of the reviewers whose scores were initially below the acceptance threshold engaged in discussions. Both indicated that they still evaluate the paper as below the threshold (one of them stated they would raise the score from 2 to 4 but not more than that).

The AC agrees with the reviewers that an analysis (theoretical or empirical) of the optimal spectral shape of the noise would significantly strengthen the paper. At the same time, the AC partially agrees with the authors that even without such an analysis, the method can be of practical value because it requires only a single parameter to tune. However, since this is a training hyper-parameter and not an inference-time one, the cost of sweeping over it is not negligible, especially if this technique is to be used for very large scale datasets. Consequently, the analogy to test-time parameters like CFG mentioned by the authors seems inaccurate. Furthermore, the new ImageNet-1k experiments use a DINOv2 latent space, which is quite different from the popular VAE spaces that are currently used by most models. Since VAEs tend to whiten the data, it is not clear whether shaping the noise spectrum provides an advantage for those popular latent diffusion models.

Overall, the AC believes the paper has merits but is not yet ready for publication in this ICLR. The authors are encouraged to address the concerns mentioned above for a future submission.

**Reviewer Concerns:**

As mentioned above, most concerns have been properly addressed by the rebuttal. The major concerns that have not been provided conclusive answers, relate to when spectral noise shaping helps, and to what extent it is expected to help for popular large-scale models (e.g. those that work in a VAE latent space).

**Reviewer Scores:**

**SWmF: score 4.**

This is also the original score. Following the rebuttal, the reviewer stated they would like to keep this score.

**muuh: score 4.**

The original score was 2 and following the rebuttal, the reviewer indicated they would like to raise the score to borderline reject.

**XYF2: score 4.**

The reviewer’s original score was 6, but the confidence level was only 2. The reviewer’s question regarding data-driven approaches to design the spectral shape was provided the answer that this was tried but didn’t help consistently. Since variants of this point were raised also by other reviewers, whose confidence was higher, and these reviewers were left unsatisfied with the answer, it is likely that reviewer XYF2 would have lowered the score to match those of the other reviewers.

**iUbf : score 4.**

The reviewer’s original score was 2. The main concerns were that the method was not exemplified on larger-scale datasets, that it doesn’t always lead to improvement, and that determining the optimal value of $\alpha$ may be computationally demanding. The authors added a larger-scale experiment and provided a convincing answer regarding the fact that when the method does not help it coincides back with the standard setting, so it never harms. For those reasons, the AC predicts that the reviewer would raise the score. However, it is likely that the answer regarding the sweep over $\alpha$ would remain a concern, and therefore the reviewer would not give a positive score.

---

### Decision · Program_Chairs · 2026-01-26

Reject